# AISciVision: A Framework for Specializing Large Multimodal Models in Scientific Image Classification

## Abstract

Trust and interpretability are crucial for the use of Artificial Intelligence (AI) in scientific research, but current models often operate as black boxes offering limited transparency and justifications for their outputs. Motivated by this problem, we introduce AISciVision, a framework that specializes Large Multimodal Models (LMMs) into interactive research partners and classification models for image classification tasks in niche scientific domains. Our framework uses two key components: (1) Visual Retrieval-Augmented Generation (VisRAG) and (2) domain-specific tools utilized in an agentic workflow. To classify a target image, AISciVision first retrieves the most similar positive and negative labeled images as context for the LMM. Then the LMM agent actively selects and applies tools to manipulate and inspect the target image over multiple rounds, refining its analysis before making a final prediction. These VisRAG and tooling components are designed to mirror the processes of domain experts, as humans often compare new data to similar examples and use specialized tools to manipulate and inspect images before arriving at a conclusion. Each inference produces both a prediction and a natural language transcript detailing the reasoning and tool usage that led to the prediction. We evaluate AISciVision on three real-world scientific image classification datasets: detecting the presence of aquaculture ponds, diseased eelgrass, and solar panels. Across these datasets, our method outperforms fully supervised models in low and full-labeled data settings. AISciVision is actively deployed in real-world use, specifically for aquaculture research, through a dedicated web application that displays and allows the expert users to converse with the transcripts. This work represents a crucial step toward AI systems that are both interpretable and effective, advancing their use in scientific research and scientific discovery.

## 1 Introduction

Until recently, meaningful interactions with AI models were largely restricted to researchers and practitioners, who accessed these models through technical interfaces, often for niche applications. But the emergence of Large Multimodal Models (LMMs) such as OpenAI's GPT (Achiam et al., 2023), Google's Gemini (Gemini-Team, 2024), and Meta's Llama (Touvron et al., 2023) has dramatically transformed the landscape. Now, both experts and the general public can converse meaningfully with AI, making these interactions a part of everyday life. This change highlights not only the rapid advancements in model capabilities but also introduces new standards for how we think about and interact with AI, as accessible, versatile, and personal assistants.

However, while this transformation has enabled AI to serve as a general-purpose assistant across a wide range of topics (Kiros et al., 2014; Vinyals et al., 2015; Ramesh et al., 2021; Abdelhamed et al., 2024), it

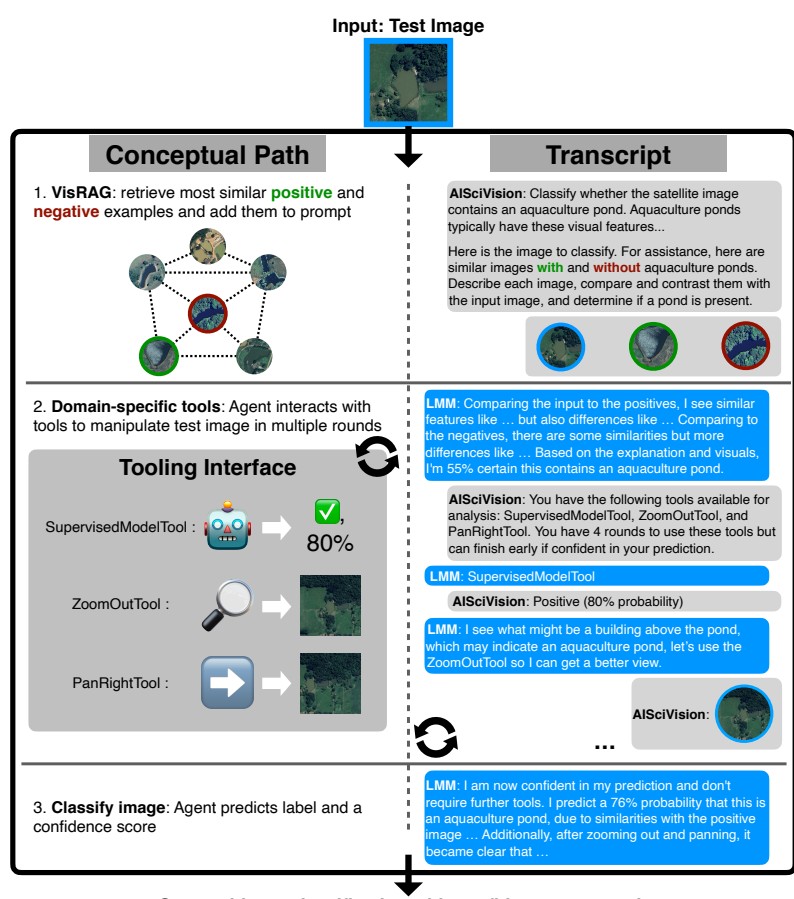

Figure 1: A schematic of our **AISciVision** framework, comprising of two components (1) Visual Retrieval-Augmented Generation (VisRAG) and (2) Domain-specific Tools. Given an input test image, we use the VisRAG component to retrieve the most similar positive and negative examples from the training dataset. We then prompt the LMM to compare these images with the test image, and utilize domain-specific interactive tools over several rounds in a conversation where the *LMM acts as an AI agent*. Finally, the agent outputs a prediction and the inference transcript. The transcript offers insight into the agent's reasoning process, improving interpretability, transparency, and trust, all crucial for applications in scientific domains.

raises a critical question: can these models provide the same depth of expertise in highly specialized and impactful domains? In areas such as medicine, law, scientific research and scientific discovery, the need goes beyond general conversations, these fields demand models capable of deep, domain-specific reasoning (Lu et al., 2022; Mall et al., 2024). The general knowledge embedded in LMMs falls short of the nuanced expertise required for these specialized tasks, limiting their effectiveness where it matters most.

Fortunately, the large context windows of LMMs allow for flexible specialization via in-context learning. By providing rich prompts and context relevant to a particular task, LMMs can adapt to domain-specific requirements, a strategy that is driving exciting research in Retrieval-Augmented Generation (RAG) (Lewis

et al., 2020; Khandelwal et al., 2020). RAG techniques enhance LMM predictions by retrieving task-specific examples, effectively refining the model's responses based on context and thereby specializing it for the task at hand. This approach is particularly valuable because the sheer scale and cost of training LMMs are often only feasible for large organizations. For most researchers, leveraging context is a practical way to harness the general knowledge of LMMs while also enabling them to excel at unique, specialized tasks (Soong et al., 2024; Zakka et al., 2024). One critical area for specialization is scientific image classification. The ability to adapt LMMs to these domains would be transformative, as it has the potential to democratize access to advanced analysis previously limited by the scarcity of domain expertise. By enabling AI to provide nuanced, explainable insights in specialized scientific tasks, we can significantly accelerate research and discovery, bridging the gap between powerful, general-purpose AI models and the pressing, domain-specific challenges faced by experts.

To address these challenges, we introduce **AISciVision**, a framework that adapts general-purpose LMMs to accurately classify scientific images while generating transparent, context-specific reasoning transcripts. AISciVision operates in a model-agnostic way, combining two key components: a Visual Retrieval-Augmented Generation system (VisRAG) and an interactive tooling AI agent that classifies images through dialogue. Users start by providing labeled training data, which is embedded into a feature space, where the system organizes positive and negative class examples separately. They also specify classification tools, ranging from basic image adjustments (e.g., contrast) to domain-specific operations like zooming into satellite imagery. During inference, AISciVision retrieves similar positive and negative image examples from the training set based on cosine similarity in the embedding space and uses them as context for the LMM's analysis. The LMM then engages in multiple rounds of interactive analysis, using the specified tools to refine its understanding of the target image, before arriving at a final prediction. A visual schematic of AISciVision is provided in Figure 1.

We evaluate our method on three real-world scientific image classification datasets: detecting aquaculture ponds in satellite images (Greenstreet et al., 2023), diseased eelgrass (Rappazzo et al., 2021), and solar panels in satellite images (Lacoste et al., 2023). Our AISciVision framework outperforms both fully supervised and zero-shot methods while producing transcripts detailing the agent's reasoning. We deploy our framework as a web application for real-time scientific monitoring, where users can interact with the inference transcripts in a 'Chat-GPT' style and can ask clarifying questions and provide corrections/feedback. Future work will study how this feedback can be incorporated into the VisRAG, to allow for the model to ever-improve and learn as experts converse with it.[1] Our contributions are:

1. We introduce **AISciVision**, a novel framework for specializing Large Multimodal Models (LMMs) to niche scientific image classification tasks. It combines Visual Retrieval-Augmented Generation (VisRAG) with domain-specific interactive tools, designed to emulate how human experts manipulate and inspect images. Through multi-round conversations using these tools, the LMM acts as an AI agent. To our knowledge, AISciVision is among the first to apply such techniques to specialized scientific applications.

2. We demonstrate the framework's efficacy on three real-world scientific image classification datasets, highlighting that our framework is flexible and easily extendable to new applications. We outperform several fully supervised models and zero-shot approaches, while additionally providing inference transcripts. These transcripts enhance interpretability, transparency, and trust in model predictions by making the entire reasoning process accessible.

3. We have deployed AISciVision as a web application that ecologists and scientists use to classify images and generate inference transcripts. The web application lays the groundwork for collecting rich and nuanced expert feedback about the LMM agent's reasoning process, opening up potential future work to incorporate such feedback to improve the agent's performance.

---

[1]The web application link will be included in the final paper.

## 2 RELATED WORKS

Our work builds on recent research in multimodal models, Retrieval-Augmented Generation, and interactive AI agents that leverage tools. We discuss and compare related work in this section, and distinguish our work as integrating these domains into a unified approach.

**Multimodal models in low-labeled data regimes**  Large Multimodal Models (LMMs) like CLIP (Radford et al., 2021), GPT-4 (Achiam et al., 2023), LLaVa (Liu et al., 2023), and PaLM-E (Driess et al., 2023) have been demonstrated to understand and generate content across multiple modalities like text, visual, and audio. By building rich and general-purpose representations of inputs from large and diverse datasets, these LMMs demonstrate competitive zero- and few-shot capabilities on a wide range of tasks (Brown et al., 2020), such as natural language understanding (Kiros et al., 2014), image captioning (Vinyals et al., 2015), text-to-image generation (Ramesh et al., 2021), and image classification (Guillaumin et al., 2010; Abdelhamed et al., 2024). Acosta et al. (2022); Moor et al. (2023); Wang et al. (2023) demonstrate that these capabilities are key for advancing research in domains where obtaining labeled data is costly and tedious, for instance, in scientific research. On the other hand, Lu et al. (2022); Mall et al. (2024) find that zero- and few-shot capabilities of general-purpose LMMs like CLIP do not suffice in scientific applications, where utilizing domain-specific information becomes important. Lu et al. (2022) find that Chain-of-Thought prompting (Wei et al., 2022) improves question-answering performance, whereas Mall et al. (2024) demonstrate that a "Vision Language Model" explicitly trained on satellite images outperforms CLIP. Our framework AISciVision extends general-purpose LMMs to classify images in low-labeled data regimes like scientific applications. We incorporate Retrieval-Augmented Generation and domain-specific tool use to ground outputs in such specialized scientific applications. AISciVision allows an LMM to predict after multiple rounds of tool use, thus going beyond classical Chain-of-Thought prompting.

**Retrieval-Augmented Generation (RAG)**  Since Large Language Models (LLMs) suffer from hallucinations in generations and static memory about the world, there is much work in Retrieval-Augmented Genereation (RAG) to retrieve relevant context from external knowledge sources to ground generations in reality (Khandelwal et al., 2020; Lewis et al., 2020). RAG has proven useful for language generation and question answering in scientific applications such as biomedical research and medicine (Soong et al., 2024; Zakka et al., 2024; Bae et al., 2024). Recent work has also demonstrated the effectiveness of RAG in multimodal settings by retrieving relevant images and documents to enrich the model's prompt, in applications of visual question answering and image captioning (Chen et al., 2022; Yasunaga et al., 2023; Lin et al., 2023). Our framework AISciVision leverages RAG for scientific image classification, by enriching the LMM's context with domain-relevant examples during inference.

**Interactive AI Agents and tool-use**  In recent years, Large Language Models have enabled users to engage in multi-turn natural language conversations to perform a wide range of tasks, from brainstorming and writing to writing code and solving math equations (Achiam et al., 2023; Driess et al., 2023; Touvron et al., 2023; Gemini-Team, 2024). There is growing interest in enabling such models to *act as an agent* on their generations, by deploying them in environments (Fan et al., 2022; Wang et al., 2024) and attaching tools to interact with the web (Nakano et al., 2022; Patil et al., 2023; Schick et al., 2023). Yao et al. (2023) accomplish this with ReAct, by prompting the model to invoke calls to specified tools in natural language format, whereas Schick et al. (2023) introduce ToolFormer which is a model finetuned on a list of available tools. When environments do not return natural language feedback, e.g. when the feedback is scalar or binary, Shinn et al. (2023) find that training a helper model to generate natural language descriptions of the feedback improves the agent's performance—termed as Reflexion. Lu et al. (2024) introduce the "AI Scientist" that imitates the research process in the Machine Learning community. They utilize Chain-of-Thought, Reflexion, and tool use to imitate the research process: brainstorm research ideas, execute experiments, and write a paper.

For scientific research in physical, life, and climate sciences, not only is it important to obtain accurate predictions but also to get insight into the underlying justifications that lead to these predictions (Wang et al., 2023; Chen et al., 2021; Society, 2024; Kong et al., 2022). With the ReAct approach of interacting with external tools and knowledge sources in a multi-turn conversation, Yao et al. (2023) shows that the agent leaves a 'paper trail' or a 'transcript' of the decision-making process. Our framework **AISciVision accomplishes exactly this feat, and is one of the first attempts at developing an interactive AI agent for scientific image classification**. Following the extensive research in tool use, we develop domain-specific tools in the AISciVision framework, for instance, zoom/pan for satellite image datasets and image enhancement tools. Our agent interacts with these tools in a multi-turn fashion and leaves a conversation transcript, enhancing interpretability, transparency, and trust—all crucial for scientific research.

## 3 METHODOLOGY

Our proposed framework, **AISciVision**, integrates a Visual Retrieval-Augmented Generation (VisRAG) procedure with domain-specific tools, which an LMM uses to classify images in the scientific domain. In this section, we describe these two components and discuss how AISciVision uses them during inference.

### 3.1 RETRIEVING RELEVANT IMAGES WITH VISUAL RAG (VISRAG)

To specialize a general-purpose LMM for scientific image classification during inference, we enrich the model's prompt with images relevant to the given test image. We first encode all available training images into a shared embedding space. Let $\mathcal{D} = \{(\boldsymbol{x}_i, y_i)\}_{i=1}^{N}$ be a training set of binary labeled images, where $\boldsymbol{x}_i \in \mathbb{R}^{H \times W \times C}$ is an image and $y_i \in \{\pm 1\}$ is its label. We map each image $\boldsymbol{x}_i$ to an embedding $\boldsymbol{e}_i = \phi(\boldsymbol{x}_i) \in \mathbb{R}^d$ using an embedding model $\phi : \mathcal{X} \to \mathbb{R}^d$ on the set of images $\mathcal{X}$. This embedding model could be a pre-trained image embedding model, for instance, CLIP (Radford et al., 2021), that ensures that similar visual content has similar embeddings. Second, we separate the embeddings into two sets: positive examples $\mathcal{E}^+ = \{\boldsymbol{e}_i \mid y_i = 1\}$ and negative examples $\mathcal{E}^- = \{\boldsymbol{e}_i \mid y_i = -1\}$. This allows us to enrich the model's prompt with structured context, described below.

On inference, we embed an input test image $\boldsymbol{x}_{\text{test}}$ to get embedding $\boldsymbol{e}_{\text{test}} = \phi(\boldsymbol{x}_{\text{test}})$. We then retrieve relevant images to enrich the LMM's context by computing the cosine similarity of the test image embedding $\boldsymbol{x}_{\text{test}}$ with all positive embeddings $\mathcal{E}^+$, and with all negative embeddings $\mathcal{E}^-$. By ranking all embeddings according to the cosine similarity, we obtain the most similar positive example $\boldsymbol{e}_{\text{sim}}^+$ and negative example $\boldsymbol{e}_{\text{sim}}^-$ as follows:

$$\boldsymbol{e}_{\text{sim}}^+ := \arg\max_{\boldsymbol{e}_i \in \mathcal{E}^+} \cos(\boldsymbol{e}_{\text{test}}, \boldsymbol{e}_i) \quad \text{and} \quad \boldsymbol{e}_{\text{sim}}^- := \arg\max_{\boldsymbol{e}_i \in \mathcal{E}^-} \cos(\boldsymbol{e}_{\text{test}}, \boldsymbol{e}_i), \quad \text{where } \cos(\boldsymbol{e}_{\text{test}}, \boldsymbol{e}_i) = \frac{\boldsymbol{e}_{\text{test}} \cdot \boldsymbol{e}_i}{\|\boldsymbol{e}_{\text{test}}\| \|\boldsymbol{e}_i\|}.$$

We then provide the images $\boldsymbol{x}_{\text{sim}}^+$ and $\boldsymbol{x}_{\text{sim}}^-$ of the respective embeddings $\boldsymbol{e}_{\text{sim}}^+$ and $\boldsymbol{e}_{\text{sim}}^-$ to the LMM in its prompt. This enables the model to evaluate images it might not have been trained on, in our case, scientific images. Adding both positive and negative examples provides relevant visual features that characterize the domain-specific classification task, effectively helping to ground the model's reasoning. Hence, this VisRAG approach facilitates more accurate and context-aware inference, leveraging the structure inherent in the classification task.

### 3.2 DOMAIN-SPECIFIC INTERACTIVE TOOLS

We leverage expert-designed tools for each classification task, empowering the LMM to refine its predictions by interacting with these tools. These tools mimic transformations and "expert advice" that a human would use to manipulate, inspect, and analyze images, before attempting to classify an image. Therefore, by

interacting with these tools in the AISciVision framework, the LMM acts akin to an interactive AI agent making informed and interpretable decisions.

Generally, we define a tool $T$ as a function on images $\mathcal{X}$, with outputs as images $\mathcal{X}$ or a real-valued scalar in $\mathbb{R}$. That is, such a tool either transforms an image $\boldsymbol{x} \in \mathcal{X}$ to another image $T(\boldsymbol{x}) \in \mathcal{X}$ or returns a numeric output $T(\boldsymbol{x}) \in \mathbb{R}$ such as a confidence score from an external model. For example, a tool $T_{\mathrm{ML}}$ might use an externally-trained Machine Learning model with parameters $\theta$ to predict the probability of a label given the image: $T_{\mathrm{ML}} = \Pr[y = 1 \mid \boldsymbol{x}; \theta]$. Other tools $T_{\mathrm{br}}$ or $T_{\mathrm{co}}$ might adjust brightness or contrast by some value $\alpha$ such that $T_{\mathrm{br}} = \mathrm{AdjustBrightness}(\boldsymbol{x}, \alpha)$ or $T_{\mathrm{co}} = \mathrm{IncreaseContrast}(\boldsymbol{x}, \alpha)$.

For each image classification task, we define a set of tools $\mathcal{T} = \{T_1, \ldots, T_K\}$ and provide their descriptions in natural language as a prompt to the LMM. At any turn $i$ in the conversation, the model submits a request for a tool $T_i \in \mathcal{T}$, which can either transform the image $\boldsymbol{x}' = T_i(\boldsymbol{x})$ or return a numeric value. The AISciVision parses this request and returns the tool's result as a prompt, with the transformed image or the numeric value described in a sentence. In essence, this is similar to ReAct (Yao et al., 2023) in that we use a hardcoded prompt template for the response. Iterative use of domain-specific tools enables the agent in our AISciVision framework to refine predictions in a context-aware manner, not only improving accuracy (see Section 4) but also producing a transcript of the agent's reasoning. Such a transcript provides interpretable insight into our framework's reasoning, which is crucial for applications in scientific discovery.

### 3.3 Inference Process in the AISciVision Framework

Given an input test image, AISciVision enriches the model's prompt with VisRAG and descriptions of the tools, which the model calls in subsequent turns of the conversation. Finally, the model outputs a classification label with a probability score indicating its confidence.

We design the initial system prompt to reflect the specified domain (see Appendix C for example transcripts). First, we use the VisRAG approach to retrieve the most similar positive and negative examples from the training set, $\boldsymbol{x}_{\mathrm{sim}}^+$ and $\boldsymbol{x}_{\mathrm{sim}}^-$ respectively. We then describe the set $\mathcal{T}$ of available domain-specific tools, and encourage the model to use them to obtain more context during inference. After a few conversation turns, the LMM responds with a binary prediction and a confidence score. The transcript of interactions represents a record of justifications at the inference stage, which a domain expert can review after the fact. In this way, AISciVision presents an interactive AI agent that uses domain-specific knowledge to not only classify scientific images but also justify its underlying reasoning in natural language.

## 4 Experiments

We extensively evaluate our AISciVision framework on three image datasets from scientific applications: detecting the presence of aquaculture ponds (Greenstreet et al., 2023), diseased eelgrass (Rappazzo et al., 2021), and solar panels (Lacoste et al., 2023). We compare against natural baselines and conduct ablation studies on components of AISciVision. We discuss our experimental results in this section.

**Datasets and Experimental Setup** We provide a brief overview of the three datasets and their significance to environmental research, along with a summary of the domain-specific tools used in the AISciVision framework (see Appendix A for a full list of tools).

1. **Aquaculture Pond Detection.** Aquaculture, vital for the global food supply, requires careful monitoring from satellite imagery. This poses challenges due to the varied appearance of water bodies. The Aquaculture dataset contains 799 images ($640 \times 640$) from Rondônia, Brazil (Greenstreet et al., 2023), with $\approx 20\%$ containing aquaculture ponds. Since the metadata includes geospatial location data, in AISciVision we define tools like zoom and pan, utilizing real-time Google Maps API.

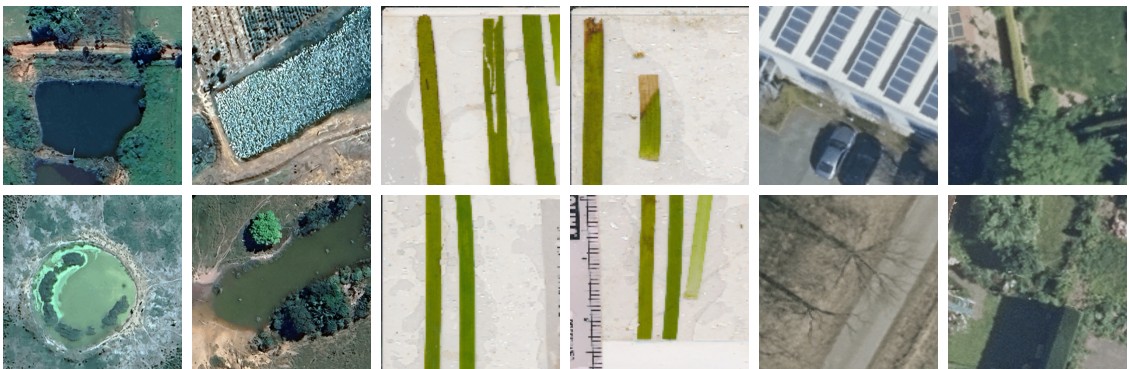

Figure 2: Four examples images from each dataset: Aquaculture, Eelgrass, and Solar (left to right). The top row are all positive examples (aquaculture ponds present, eelgrass plant diseased, and solar panels present), and the bottom row are all negative examples.

2. **Eelgrass Wasting Disease Detection.** Eelgrass (*Zostera marina*), essential for coastal ecosystems, faces threats from Eelgrass Wasting Disease (EWD) (Groner et al., 2016). The Eelgrass dataset contains 9887 images ($128 \times 128$) from Washington state, with $\approx 45\%$ showing diseased eelgrass (Rappazzo et al., 2021). We incorporate tools like contrast and sharpening adjustments.

3. **Solar Panel Detection.** The open-source Solar dataset tracks solar panel adoption with satellite images (Lacoste et al., 2023). It contains 11814 images ($320 \times 320$), with $\approx 15\%$ containing solar panels. We use similar image enhancement tools as the Eelgrass dataset, since geospatial metadata is absent.

We test on 100 randomly subsampled examples from each dataset's test set for consistent evaluation across all methods. To balance the cost constraints of LMM experimentation, we use this small test set for robust experiments and ablation studies. All methods are evaluated in low-labeled (20%) and full-labeled (100%) data settings, on Accuracy, F1-score, and Area Under Curve (AUC) metrics.

**AISciVision method**    We use the GPT-4o as the framework's LMM, and attach the two components Vis-RAG and domain-specific tools for each dataset. We provide text descriptions of the available tools as prompts to the LMM, and instruct how to request tool use. We encourage the LMM to use at least 3 tools during inference, and instruct it to submit a classification decision with a confidence value in 4 conversation turns. All embeddings for VisRAG are computed via CLIP, and the cosine similarity is used to gauge the similarity between images. We include example transcripts and our prompts for all datasets in Appendix C.

**Baselines**    A key component of AISciVision is VisRAG, which retrieves the most similar positive and negative examples during inference before prompting the LMM. Hence, a natural baseline is naïve $k$-Nearest Neighbor ($k$-**NN** with $k = 3$) using CLIP embeddings. We also include **CLIP-ZeroShot** as a baseline: we classify a test image by comparing the cosine similarities of CLIP text embeddings of the two labels with the CLIP image embedding. As CLIP-ZeroShot does not rely on image features specific to the scientific domain, we evaluate another baseline **CLIP+MLP** for binary classification, where we train a 2-layer Multi-Layer Perceptron (MLP) on top of frozen CLIP image embeddings. Additionally we train a Resnet50 model (He et al., 2016) as a fully supervised baseline (see Appendix F for training details). We find that

| Methods | Aquaculture | | | | | | Eelgrass | | | | | | Solar | | | | | | Transcript |
|---|---|---|---|---|---|---|---|---|---|---|---|---|---|---|---|---|---|---|---|
| | 20% | | | 100% | | | 20% | | | 100% | | | 20% | | | 100% | | | |
| | Acc | F1 | AUC | Acc | F1 | AUC | Acc | F1 | AUC | Acc | F1 | AUC | Acc | F1 | AUC | Acc | F1 | AUC | |
| $k$-NN | 0.84 | 0.65 | 0.83 | 0.80 | 0.52 | 0.76 | 0.71 | 0.66 | 0.79 | 0.78 | 0.72 | 0.86 | 0.96 | 0.80 | 0.91 | 0.95 | 0.76 | 0.94 | ✗ |
| CLIP-ZeroShot | 0.82 | 0.00 | 0.56 | 0.82 | 0.00 | 0.56 | 0.60 | 0.17 | 0.54 | 0.60 | 0.17 | 0.54 | 0.63 | 0.33 | 0.65 | 0.63 | 0.33 | 0.65 | ✗ |
| CLIP+MLP | 0.85 | 0.67 | 0.91 | 0.86 | 0.63 | **0.93** | 0.82 | 0.72 | 0.88 | 0.80 | 0.74 | 0.92 | 0.98 | 0.91 | 1.00 | 0.99 | 0.96 | 1.00 | ✗ |
| Resnet50 | 0.78 | 0.59 | 0.79 | 0.82 | 0.40 | 0.62 | **0.87** | **0.86** | 0.87 | **0.94** | **0.93** | 0.94 | 0.98 | 0.91 | 0.92 | 0.97 | 0.86 | 0.88 | ✗ |
| AISciVision | **0.90** | **0.78** | **0.95** | **0.92** | **0.81** | **0.93** | 0.81 | 0.73 | **0.89** | 0.84 | 0.80 | **0.95** | **0.98** | **0.92** | **1.00** | 0.97 | 0.88 | **1.00** | ✓ |

Table 1: Our AISciVision framework (GPT4o + VisRAG + Tools) consistently outperforms zero-shot and supervised methods across all datasets and metrics. We evaluate all methods in both low-labeled (20%) and full-labeled (100%) regimes. The CLIP+MLP supervised model, trained on the same images, is used as a tool within AISciVision, providing prediction probabilities upon request by the LMM. We observe that the naïve $k$-NN baseline outperforms CLIP-ZeroShot and is competitive with CLIP+MLP. This highlights the importance of leveraging domain-specific structure in scientific image classification over general-purpose LMMs, justifying the VisRAG component in AISciVision. We further find that AISciVision outperforms CLIP+MLP, which is provided as a tool in our framework, implying that AISciVision does not solely rely on this tool. Further, we trained a Resnet50 model and found that AISciVision consistently outperformed. ResNet-50's performance was hindered by class imbalances, with more training data amplifying bias toward the majority class. Precision and recall are detailed in Table 3.

AISciVision consistently outperforms all baselines on all three datasets, in both low- and full-labeled data regimes (Table 1). [2]

**Ablation studies and tool efficacy**   We conduct ablation studies on the two components of AISciVision: VisRAG and domain-specific tools. We compare the following variants with AISciVision (GPT4o + VisRAG + Tools): (1) **GPT4o-ZeroShot** that predicts a label and confidence score after the initial prompt[3], (2) **GPT4o + VisRAG** that uses only VisRAG to retrieve examples relevant to the test image, (3) **GPT4o + Tools** that uses only our domain-specific tools. We report these ablation experiment results in Table 2. These ablation experiments allow us to isolate and evaluate the effects of retrieval through VisRAG and usage of domain-specific tools.

Moreover, we log all inference transcripts and frequencies of tool usage in AISciVision. We then analyze the effect of different tools on the agent's classification accuracy, reported in Figure 3, offering insight into their efficacy for the agent's decision-making process.

## 5 DEPLOYED APPLICATION

AISciVision is not just a conceptual framework, we have deployed it as a fully functional web application to detect aquaculture ponds, enabling real-time and scalable use by ecologists (see a screenshot of the interface in Figure 4). When ecologists upload a test image to the web application, the AISciVision framework prompts the LMM to detect if the image contains aquaculture ponds. Importantly, AISciVision provides the detailed transcript that outlines the LMM's reasoning and tool use. AISciVision's deployed interface not only provides an accurate classification result but also a transparent decision-making process—crucial for expert ecologists to validate the framework.

---

[2]In Table 1, note that CLIP-ZeroShot does not use any training images from the datasets, hence the same values of metrics between the low- and full-labeled data regimes. In our experiments, CLIP-ZeroShot does indeed attain 0.0 F1 score for the Aquaculture dataset, because it does not predict any true positives (aquaculture ponds present). It has been shown that CLIP embeddings perform poorly zero-shot on satellite images (Radford et al., 2021; Mall et al., 2024).

[3]GPT4o generations are inherently random even after setting a seed, system fingerprint, the temperature to 0. This leads to slight variability in the values of metrics.

| Methods | Aquaculture | | | | | | Eelgrass | | | | | | Solar | | | | | |
|---|---|---|---|---|---|---|---|---|---|---|---|---|---|---|---|---|---|---|
| | 20% | | | 100% | | | 20% | | | 100% | | | 20% | | | 100% | | |
| | Acc | F1 | AUC | Acc | F1 | AUC | Acc | F1 | AUC | Acc | F1 | AUC | Acc | F1 | AUC | Acc | F1 | AUC |
| GPT4o-ZeroShot | 0.86 | 0.67 | 0.85 | 0.85 | 0.62 | 0.86 | 0.77 | **0.74** | 0.86 | 0.74 | 0.71 | 0.83 | 0.93 | 0.77 | 0.99 | 0.91 | 0.73 | 0.99 |
| GPT4o + VisRAG | 0.87 | 0.68 | 0.89 | 0.85 | 0.67 | 0.90 | 0.77 | **0.74** | 0.86 | **0.86** | **0.83** | 0.92 | 0.97 | 0.89 | 0.99 | **0.97** | **0.89** | 0.99 |
| GPT4o + Tools | 0.88 | 0.74 | 0.88 | 0.90 | 0.76 | 0.91 | 0.79 | **0.74** | **0.90** | 0.77 | 0.74 | 0.90 | 0.96 | 0.86 | **1.00** | 0.96 | 0.86 | **1.00** |
| AISciVision | **0.90** | **0.78** | **0.95** | **0.92** | **0.81** | **0.93** | **0.81** | 0.73 | 0.89 | 0.84 | 0.80 | **0.95** | **0.98** | **0.92** | **1.00** | **0.97** | 0.88 | **1.00** |

Table 2: We conduct ablation studies on each component of our framework, AISciVision (GPT-4o + Vis-RAG + Tools), and find that it generally outperforms the ablation methods. Even so, the isolated benefits of VisRAG and Tools are notable in Eelgrass and Solar datasets. While geospatial tools are particularly effective in the Aquaculture dataset of satellite images, the benefits of VisRAG complements those of domain-specific tools to significantly outperform either component separately. On a closer look at inference transcripts of the Eelgrass and Solar datasets, we observe that the supervised model tool can sometimes introduce a bias that degrades AISciVision's performance (see transcripts in Appendix D). We leave it to future work to improve upon our image enhancement tools and develop techniques to select the optimal toolset.

This practical prototype opens the potential for expert ecologists to return feedback on both the classification results and the underlying reasoning. We aim to integrate this feedback algorithmically to improve the VisRAG component of AISciVision, building a framework that improves with expert use. There are many exciting research directions: seamlessly incorporating feedback loops, utilizing enhanced tools and understanding their interactions, and improving the framework's adaptability to other scientific applications.

## 6 DISCUSSION

Our AISciVision framework combines robust prediction capabilities, transparency, and adaptability, offering a practical approach for AI use in scientific contexts. By delivering accurate and context-aware predictions along with a full reasoning transcript, AISciVision serves as a valuable research partner across different applications. Transcripts enhance the accountability and traceability of model outputs, enabling researchers

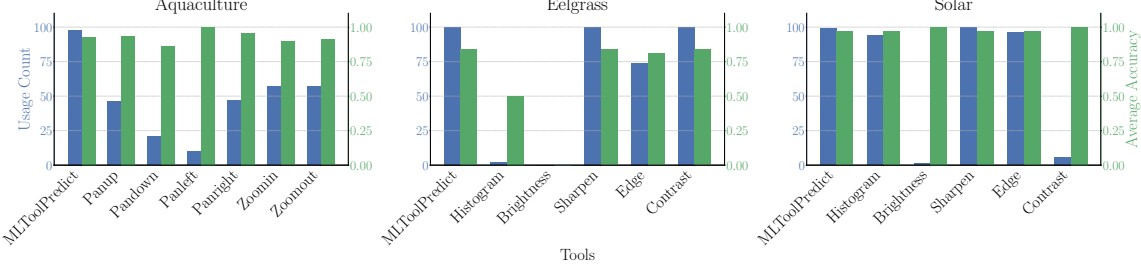

Figure 3: We analyze the frequency of tool use in AISciVision for each dataset, and their effects on the final classification result. **Blue** bars indicate the number of tool calls during inference on our test sets of 100 images, whereas **green** bars indicate the mean classification accuracy *when* the specified tool was called in a conversation in *any* round. Note that the LMM can call a tool multiple times in the same conversation. Across all datasets, the LMM almost always requests the supervised model tool "MLToolPredict" but does not solely rely on the returned results. The LMM heavily relies on geospatial tools for the Aquaculture dataset, since these tools return additional information from the vicinity. It is curious that the LMM does not use the AdjustBrightness tool at all and uses HistogramEqualization sparingly for the other two datasets. All tools are described in Appendix A.

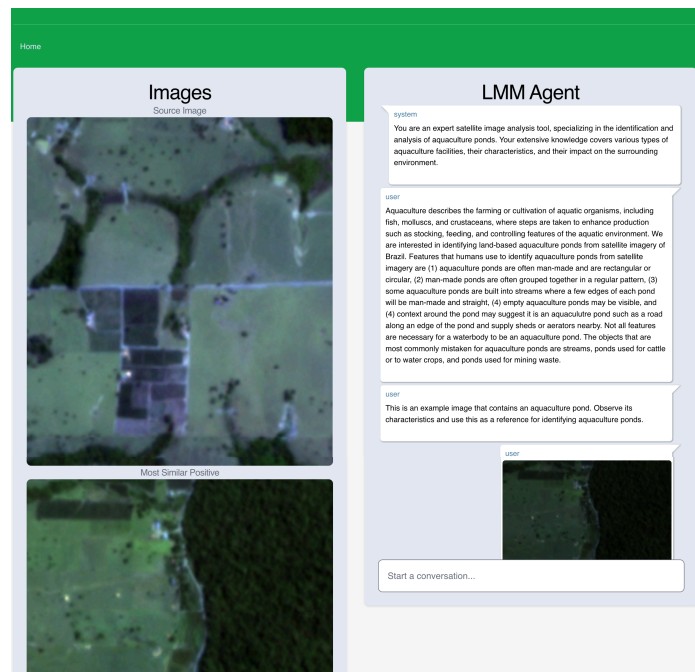

Figure 4: Screenshot of the AISciVision web application for aquaculture pond detection. The application allows users to upload satellite images, interact with visual tools such as zoom and contrast adjustment, and view predictions alongside retrieved similar examples. Additionally, the produced transcript is conversable in a 'Chat-GPT' like manner. Allowing the user to ask questions about the inference steps, or to even give rich natural language feedback about errors the model might have made. For future work, we aim to incorporate this feedback into the VisRAG, enabling an ever-learning RAG system that continually improves our model.

to validate decision-making processes. This is particularly important in scientific research, for instance in conservation efforts in complex ecosystems like the Amazon Basin, where accurate spatial data is crucial for sustainability. The combination of predictions and interpretable transcripts can help advance scientific discovery, support regulatory compliance by providing a clear record of inferences, and aid education by giving concrete examples of classification processes.

**Limitations and Future Work**     While AISciVision offers significant benefits in providing transparent reasoning and the potential for enhanced accuracy, it comes with a trade-off: using off-the-shelf LMMs for inference is financially expensive, compared to traditional machine learning methods. For our future work, we aim to actively develop our web application to continue to collect expert feedback on the LMM agent's reasoning through a ChatGPT-style interface. Experts can provide real-time feedback and corrections, which will be stored within the VisRAG component to improve the LMM agent's performance with use. We envision the system continuing to learn as experts interact with the agent, and provide rich natural language feedback. Beyond refining our approach for image data, we also plan to test and extend our method to other modalities, such as sound or any tokenizable input that can be incorporated into an LMM.

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

## A    DOMAIN-SPECIFIC TOOLS

The tools developed for each dataset are designed to mimic the strategies that human experts would use when performing these specific classification tasks. For the Eelgrass wasting disease and Solar panel detection tasks, the tools focus on image enhancement techniques that a human might naturally apply when examining an image closely. These include adjusting brightness, contrast, and sharpness, as well as applying edge detection and histogram equalization. Such manipulations can reveal features or patterns that might be helpful for accurate classification, much like how a human expert might squint, tilt their head, or adjust lighting to better perceive important details.

In contrast, the aquaculture pond detection task presents a unique challenge where the presence of water bodies isn't always immediately apparent from a single satellite image. To address this, the tools for this task are designed to emulate how a human would interact with digital maps. They include options to pan and zoom, allowing for exploration of the surrounding area and changes in perspective. This approach mirrors how a human expert might navigate digital mapping software, zooming out to get context from the broader landscape, then zooming in on areas of interest to confirm the presence of aquaculture ponds. By providing these map navigation tools, the framework can gather additional spatial information that may be crucial for accurate classification, especially in cases where a single image view might be ambiguous or inconclusive.

We detail the full list of tools for each dataset in our AISciVision framework.

### A.1    TOOLS FOR AQUACULTURE DATASET

- **PredictAquaculturePondTool**: Predicts the probability of an aquaculture pond being present in the image using a machine learning model. This tool is particularly helpful when you need a quantitative assessment of the likelihood of aquaculture pond presence in the satellite image.
- **PanUpToolRelative**: Pans the view upwards relative to the last image seen.
- **PanUpToolAbsolute**: Pans the view upwards relative to the original starting image.
- **PanDownToolRelative**: Pans the view downwards relative to the last image seen.
- **PanDownToolAbsolute**: Pans the view downwards relative to the original starting image.
- **PanLeftToolRelative**: Pans the view to the left relative to the last image seen.
- **PanLeftToolAbsolute**: Pans the view to the left relative to the original starting image.
- **PanRightToolRelative**: Pans the view to the right relative to the last image seen.
- **PanRightToolAbsolute**: Pans the view to the right relative to the original starting image.
- **ZoomInToolRelative**: Zooms in on the center of the current view relative to the last image seen.
- **ZoomInToolAbsolute**: Zooms in on the center of the original view relative to the original starting image.
- **ZoomOutToolRelative**: Zooms out from the current view relative to the last image seen.
- **ZoomOutToolAbsolute**: Zooms out from the original view relative to the original starting image.

### A.2    TOOLS FOR EELGRASS DATASET

- **AdjustBrightnessTool**: Adjusts the brightness of the image by 50%. This tool can help when the image is too dark or too bright, allowing for better visibility of disease symptoms on the eelgrass blade.
- **SharpenTool**: Sharpens the image to enhance edges and details. This tool is useful for making subtle features more prominent, which can help in identifying signs of eelgrass wasting disease.

- **EdgeDetectionTool**: Applies edge detection to the image, highlighting boundaries and features. This can help in identifying lesions or patterns associated with eelgrass wasting disease.

- **IncreaseContrastTool**: Increases the contrast of the image by 50%. This tool can be helpful when the image appears too flat or when you need to enhance the visibility of subtle details, especially in cases where eelgrass wasting disease symptoms might be hard to distinguish.

- **DecreaseContrastTool**: Decreases the contrast of the image by 50%. This tool can be useful when the image appears too harsh or when you want to reduce the intensity of bright areas, which might help in identifying overall patterns or structures in the eelgrass.

- **PredictEelgrassWastingDiseaseTool**: Predicts the probability of eelgrass wasting disease in the image using a machine learning model. This tool is particularly helpful when you need a quantitative assessment of the likelihood of disease presence in the eelgrass sample.

- **HistogramEqualizationTool**: Enhances the contrast of the image using histogram equalization. This can help in making features more distinguishable, which is beneficial for detecting eelgrass wasting disease symptoms.

## A.3 TOOLS FOR SOLAR DATASET

- **HistogramEqualizationTool**: Enhances the contrast of the image using histogram equalization. This can help in making features more distinguishable, which is beneficial for detecting solar panels and potential defects.

- **AdjustBrightnessTool**: Adjusts the brightness of the image by 50%. This tool can help when the image is too dark or too bright, allowing for better visibility of solar panels and their features.

- **SharpenTool**: Sharpens the image to enhance edges and details. This tool is useful for making subtle features more prominent, which can help in identifying solar panels and potential defects.

- **EdgeDetectionTool**: Applies edge detection to the image, highlighting boundaries and features. This can help in identifying the outlines of solar panels and potential defects or anomalies.

- **IncreaseContrastTool**: Increases the contrast of the image by 50%. This tool can be helpful when the image appears too flat or when you need to enhance the visibility of subtle details, especially in cases where solar panels might be hard to distinguish from their surroundings.

- **DecreaseContrastTool**: Decreases the contrast of the image by 50%. This tool can be useful when the image appears too harsh or when you want to reduce the intensity of bright areas, which might help in identifying overall patterns or structures in the solar panel array.

- **PredictSolarPanelTool**: Predicts the probability of a solar panel being in the image using a machine learning model. This tool is particularly helpful when you need a quantitative assessment of the likelihood of a solar panel being present in the image.

## B    PRECISION AND RECALL RESULTS

In addition to the metrics reported in the main paper in Table 1, we report the Precision and Recall metrics in Table 3.

| Methods | Aquaculture | | | | Eelgrass | | | | Solar | | | |
| | 20% | | 100% | | 20% | | 100% | | 20% | | 100% | |
| | Prec | Rec | Prec | Rec | Prec | Rec | Prec | Rec | Prec | Rec | Prec | Rec |
|---|---|---|---|---|---|---|---|---|---|---|---|---|
| $k$-NN | 0.68 | 0.62 | 0.61 | 0.46 | 0.58 | 0.76 | 0.67 | 0.78 | **1.00** | 0.67 | 0.89 | 0.67 |
| CLIP-ZeroShot | 0.00 | 0.00 | 0.00 | 0.00 | 0.57 | 0.10 | 0.57 | 0.10 | 0.25 | 0.47 | 0.25 | 0.47 |
| CLIP+MLP | 0.71 | 0.62 | 0.86 | 0.50 | **0.85** | 0.62 | 0.71 | 0.78 | **1.00** | 0.83 | **1.00** | 0.92 |
| GPT4o-ZeroShot | 0.78 | 0.58 | 0.80 | 0.50 | 0.63 | **0.89** | 0.60 | 0.86 | 0.63 | **1.00** | 0.57 | **1.00** |
| GPT4o + VisRAG | **0.82** | 0.58 | 0.71 | 0.62 | 0.63 | **0.89** | **0.76** | **0.92** | 0.80 | **1.00** | 0.80 | **1.00** |
| GPT4o + Tools | 0.77 | 0.71 | 0.89 | 0.67 | 0.68 | 0.81 | 0.64 | 0.86 | 0.75 | **1.00** | 0.75 | **1.00** |
| AISciVision | **0.82** | **0.75** | **0.94** | **0.71** | 0.76 | 0.70 | 0.73 | 0.89 | 0.92 | 0.92 | 0.85 | 0.92 |

Table 3: We report values of precision and recall when all methods are tested in both low- and full-labeled training data regimes, that is 20% and 100% respectively. Recall that our framework AISciVision is GPT4o + VisRAG + Tools. CLIP-ZeroShot does not obtain any true positives for the aquaculture dataset, i.e. it only predicts that no pond is present, resulting in 0.0 values of precision, recall, and F1-score.

## C    EXAMPLE TRANSCRIPTS

In this section we provide the full transcript for one sample inference from each of the datasets. These contain the full prompts and inference steps for the AISciVision framework.

## C.1 AQUACULTURE

True Class: 1, LMM Prediction: 1 (Probability: 0.80), Supervised Prediction: 0 (Probability: 25.91)

**System:**

You are an expert in satellite imagery analysis specializing in aquaculture pond detection. Your task is to determine whether a satellite image contains aquaculture ponds or not. Aquaculture ponds in satellite imagery typically appear as regular, geometric shapes with distinct water surfaces. They are often arranged in clusters or grids in coastal areas or inland water bodies. Key characteristics to look for include: 1. Geometric Patterns: Aquaculture ponds are usually rectangular or square in shape, forming distinct geometric patterns. 2. Water Color: The water in aquaculture ponds often has a different color or tone compared to natural water bodies, due to the high density of organisms and potential algae growth. 3. Surrounding Features: Look for access roads, feeding platforms, or other infrastructure associated with aquaculture operations. 4. Location: Aquaculture ponds are commonly found in coastal areas, estuaries, or near rivers and lakes. 5. Size and Scale: The size of aquaculture ponds can vary, but they are typically uniform within a single farm and arranged in an organized manner. 6. Texture: The water surface of aquaculture ponds often appears smoother than natural water bodies. Use your knowledge and the tools provided to make an accurate classification of whether aquaculture ponds are present in the given satellite image.

**User:**

This is an example of a satellite image with an aquaculture pond. Describe what you see, noting the characteristics that identify it as an aquaculture pond.

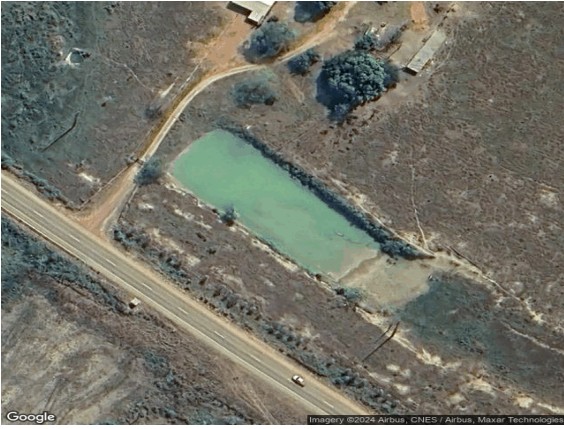

**Assistant:**

In the provided satellite image, several characteristics identify the presence of an aquaculture pond:

**Geometric Pattern**: The pond is rectangular in shape, which is a common characteristic of aquaculture ponds. The straight edges and right angles suggest human-made construction.

**Water Color**: The water in the pond has a distinct greenish hue, which is different from the surrounding natural landscape. This color difference can be due to the high density of organisms or algae growth typical in aquaculture ponds.

**Surrounding Features**: There are access roads leading to the pond, indicating human activity and maintenance. Additionally, there are structures nearby that could be related to the aquaculture operation, such as storage or feeding facilities.

**Location**: The pond is situated in a relatively open area, which is typical for aquaculture ponds to allow easy access and management.

**Size and Scale**: The pond is uniform in size and appears to be part of a managed system, which is consistent with the organized nature of aquaculture farms.

**Texture**: The water surface appears relatively smooth compared to natural water bodies, indicating controlled conditions typical of aquaculture ponds.

These characteristics collectively suggest that the rectangular water body in the image is an aquaculture pond.

**User:**

This is an example of a satellite image without an aquaculture pond. Describe what you see, noting the absence of aquaculture ponds and any other relevant features.

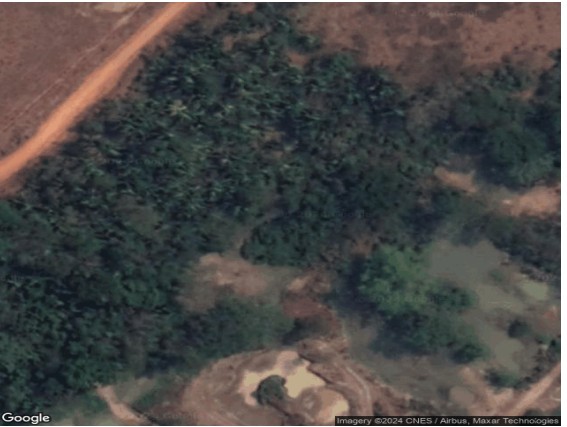

**Assistant:**

In the provided satellite image, there are no visible aquaculture ponds. Here are the observations and relevant features:

**Absence of Geometric Patterns**: There are no rectangular or square water bodies that would indicate the presence of aquaculture ponds. The area appears more natural and irregular in shape.

**Vegetation**: The image shows a dense area of vegetation, likely a forest or a plantation. The green canopy covers a significant portion of the image, indicating a natural or agricultural landscape rather than an aquaculture setup.

**Natural Water Bodies**: There are some patches of water visible, but they are irregular in shape and blend into the natural landscape. These do not exhibit the distinct, controlled appearance of aquaculture ponds.

**Surrounding Features**: There is a dirt road visible in the upper left corner of the image, but there are no signs of infrastructure typically associated with aquaculture operations, such as feeding platforms or storage facilities.

**Location and Land Use**: The area appears to be more rural and natural, with no clear signs of organized aquaculture activities. The land use seems to be more aligned with forestry or agriculture.

**Texture and Water Surface**: The water surfaces visible in the image do not have the smooth, controlled texture typical of aquaculture ponds. They appear more natural and varied.

Overall, the absence of geometric water bodies, the presence of dense vegetation, and the lack of aquaculture-related infrastructure indicate that this image does not contain aquaculture ponds.

**User:**

Here is the satellite image we need to classify as having an aquaculture pond or not. Describe what you see and compare and contrast it with the previous two known examples. Explain your thought process for classifying this image.

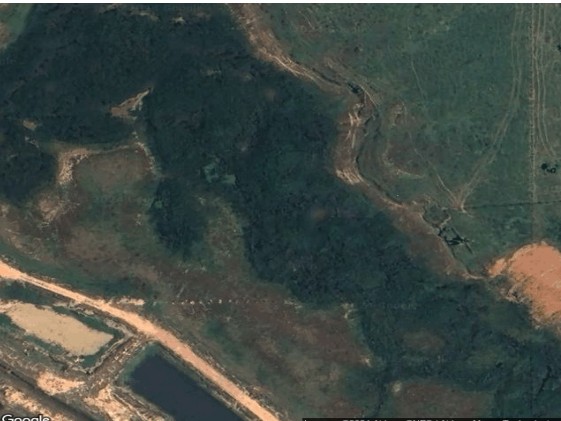

**Assistant:**

In the provided satellite image, we need to determine whether there are aquaculture ponds present. Here is a detailed analysis and comparison with the previous examples:

**Observations:**

1. **Geometric Patterns**: In the lower left corner of the image, there are rectangular water bodies that exhibit straight edges and right angles, which are indicative of human-made structures.
2. **Water Color**: The water in these rectangular areas has a distinct color, different from the surrounding natural landscape, suggesting controlled water bodies.
3. **Surrounding Features**: There are visible access paths and possibly some infrastructure around the rectangular water bodies, which could be related to aquaculture operations.
4. **Location**: The rectangular water bodies are situated in an area that appears to be managed and possibly used for aquaculture, given the presence of infrastructure and access paths.
5. **Size and Scale**: The water bodies are uniform in size and shape, consistent with the organized nature of aquaculture ponds.
6. **Texture**: The water surfaces in the rectangular areas appear smooth, which is typical of controlled aquaculture ponds.

**Comparison with Previous Examples:**

- **Example with Aquaculture Pond**: The first example showed a clear rectangular pond with distinct water color and surrounding infrastructure. The current image has similar rectangular water bodies with distinct colors and surrounding features, suggesting aquaculture ponds.
- **Example without Aquaculture Pond**: The second example showed a natural landscape with dense vegetation and irregular water bodies. The current image, in contrast, has geometric water bodies and infrastructure, which are not present in the second example.

**Thought Process for Classification:**

1. **Geometric Shape**: The presence of rectangular water bodies with straight edges strongly suggests human-made ponds.
2. **Water Color and Texture**: The distinct color and smooth texture of the water bodies are consistent with aquaculture ponds.
3. **Surrounding Infrastructure**: The visible access paths and possible infrastructure support the idea of managed aquaculture operations.
4. **Comparison with Known Examples**: The current image shares more characteristics with the first example (with aquaculture pond) than the second example (without aquaculture pond).

**Conclusion:**

Based on the geometric patterns, water color, surrounding features, and comparison with known examples, it is reasonable to classify this image as having aquaculture ponds. The rectangular water bodies with distinct characteristics and surrounding infrastructure are indicative of aquaculture operations.

**User:**

You have access to the following tools to assist in your analysis: - PredictAquaculturePondTool: Tool Name: 'PredictAquaculturePondTool' Description: Predicts the probability of an aquaculture pond being present in the image using a machine learning model. This tool is particularly helpful when you need a quantitative assessment of the likelihood of aquaculture pond presence in the satellite image. The model has been trained on various satellite images of aquaculture ponds and provides a percentage probability. However, please note that the model can be wrong or inaccurate, especially in complex or ambiguous cases. It's important to use this tool's output as one piece of evidence among others, and not rely on it exclusively for your final decision. - PanUpToolRelative: Pans the view upwards relative to the last image seen. - PanUpToolAbsolute: Pans the view upwards relative to the original starting image. - PanDownToolRelative: Pans the view downwards relative to the last image seen. - PanDownToolAbsolute: Pans the view downwards relative to the original starting image. - PanLeftToolRelative: Pans the view to the left relative to the last image seen. - PanLeftToolAbsolute: Pans the view to the left relative to the original starting image. - PanRightToolRelative: Pans the view to the right relative to the last image seen. - PanRightToolAbsolute: Pans the view to the right relative to the original starting image. - ZoomInToolRelative: Zooms in on the center of the current view relative to the last image seen. - ZoomInToolAbsolute: Zooms in on the center of the original view relative to the original starting image. - ZoomOutToolRelative: Zooms out from the current view relative to the last image seen. - ZoomOutToolAbsolute: Zooms out from the original view relative to the original starting image.

You have 4 opportunities to use these tools. You are encouraged to use at least 3 tools to gather sufficient information. While it's good to conclude early if you're certain about the classification, the most important thing is making the correct prediction. Feel free to use more tools if you need to increase your confidence.

If you want to use a tool, explain how it will assist your analysis, then respond with the tool name in square brackets, like this: [ToolName], or if you are finished, respond with [Finished].

Remember, your goal is to accurately classify whether the image shows aquaculture ponds or not. If you do use a tool, you will see the result of the tool immediately, and you should briefly, 1-2 sentences describe any new insights from the tool, and your thoughts on the classification.

After each tool use, end your thought with brackets in the format {Yes:yesPercent,No:noPercent} to indicate your current confidence/prediction. For example, {Yes:70,No:30} would indicate 70% confidence in the presence of aquaculture ponds and 30% confidence in their absence. Note that this is not your final prediction.

Then you will be asked again if you want to use a tool or are finished.

**Assistant:**

To start, I will use the PredictAquaculturePondTool to get a quantitative assessment of the likelihood of aquaculture ponds being present in the image. This will provide a baseline probability to inform my analysis.

[PredictAquaculturePondTool]

**User:**

The probability of an aquaculture pond being present in this image is 25.91%. (Percentage out of 100%) Please note that this model has been trained on a limited dataset and its predictions can be inaccurate. Use this information as a guideline rather than a definitive prediction.

You have 3 rounds left. You may now choose another tool or indicate you're finished. Respond with ONLY [ToolName] to use a single tool, or [Finished] if you're done. Do not include multiple tool names or any other text in brackets. If you want to reference the last tool you used, simply write its name without brackets. After choosing a tool or indicating you're finished, provide your confidence in the presence of aquaculture ponds in the format {yes:PercentYes,no:PercentNo}. Then, explain why you are picking this tool or finishing, and how it relates to your current assessment of the image.

**Assistant:**

[ZoomInToolRelative]

{yes:60,no:40}

I am choosing to zoom in on the current view to get a closer look at the rectangular water bodies in the lower left corner. This will help me better assess their characteristics and determine if they are indeed aquaculture ponds.

**User:**

The view has been zoomed in, providing a closer look at the current view. New zoom level: 20

You have 2 rounds left. You may now choose another tool or indicate you're finished. Respond with ONLY [ToolName] to use a single tool, or [Finished] if you're done. Do not include multiple tool names or any other text in brackets. If you want to reference the last tool you used, simply write its name without brackets. After choosing a tool or indicating you're finished, provide your confidence in the presence of aquaculture ponds in the format {yes:PercentYes,no:PercentNo}. Then, explain why you are picking this tool or finishing, and how it relates to your current assessment of the image.

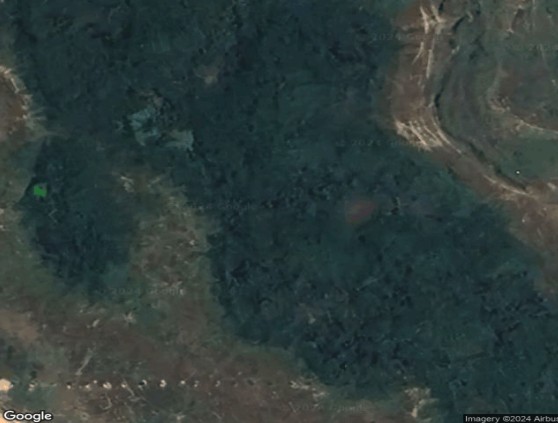

**Assistant:**

[PanDownToolRelative]

{yes:60,no:40}

I am choosing to pan down relative to the current view to get a better look at the rectangular water bodies that were partially visible in the lower left corner of the original image. This will help me confirm if they exhibit more characteristics of aquaculture ponds.

**User:**

The view has been panned down, revealing more of the area down of the previous view. New coordinates: Lat -10.337789827191068, Lon -62.823227003321655

You have 1 rounds left. You may now choose another tool or indicate you're finished. Respond with ONLY [ToolName] to use a single tool, or [Finished] if you're done. Do not include multiple tool names or any other text in brackets. If you want to reference the last tool you used, simply write its name without brackets. After choosing a tool or indicating you're finished, provide your confidence in the presence of aquaculture ponds in the format {yes:PercentYes,no:PercentNo}. Then, explain why you are picking this tool or finishing, and how it relates to your current assessment of the image.

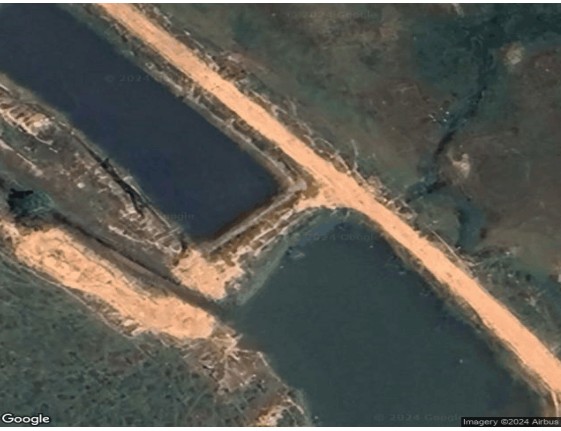

**Assistant:**

[ZoomInToolRelative]

{yes:80,no:20}

I am choosing to zoom in further on the current view to get a closer look at the rectangular water bodies. This will help me confirm the presence of aquaculture ponds by examining the details of the water bodies and surrounding infrastructure more closely.

**User:**

The view has been zoomed in, providing a closer look at the current view. New zoom level: 21

You have 0 rounds left. You may now choose another tool or indicate you're finished. Respond with ONLY [ToolName] to use a single tool, or [Finished] if you're done. Do not include multiple tool names or any other text in brackets. If you want to reference the last tool you used, simply write its name without brackets. After choosing a tool or indicating you're finished, provide your confidence in the presence of aquaculture ponds in the format {yes:PercentYes,no:PercentNo}. Then, explain why you are picking this tool or finishing, and how it relates to your current assessment of the image.

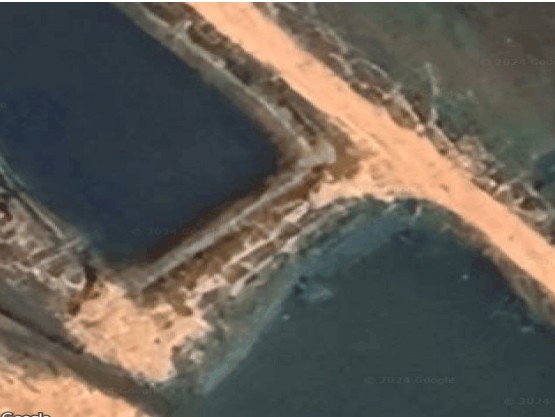

**User:**

Based on your analysis and the tools used, please provide your final classification regarding the presence of an aquaculture pond in the satellite image. Provide a brief explanation for your decision, including key observations and any tool results that influenced your conclusion. Consider the shape, color, and arrangement of water bodies, as well as any surrounding infrastructure that might indicate aquaculture activity. At the end of your response, you must provide your classification in the following format: [Yes:PercentYes,No:PercentNo], where PercentYes is the probability you think it is an aquaculture pond, and PercentNo is the probability you think it is not an aquaculture pond. These percentages must sum to 100%. To break ties, always assign at least 51% to one category. For example, [Yes:75,No:25] indicates 75% certainty of an aquaculture pond's presence, while [Yes:49,No:51] indicates 51% certainty of its absence.

**Assistant:**

Based on the analysis and the tools used, I conclude that the satellite image does show the presence of aquaculture ponds. Here are the key observations and tool results that influenced my decision:

1. **Geometric Patterns**: The water bodies in the image are rectangular with straight edges and right angles, which are indicative of human-made structures typical of aquaculture ponds.
2. **Water Color**: The water in these rectangular areas has a distinct color, different from the surrounding natural landscape, suggesting controlled water bodies.
3. **Surrounding Infrastructure**: There are visible access paths and infrastructure around the rectangular water bodies, which are consistent with aquaculture operations.
4. **PredictAquaculturePondTool**: Although the tool predicted a 25.91% probability of an aquaculture pond being present, this model's prediction is not definitive and should be used as a guideline. The visual evidence from the zoomed-in and panned views provides stronger support for the presence of aquaculture ponds.
5. **Zoom and Pan Tools**: The closer inspection provided by zooming in and panning down revealed more details of the rectangular water bodies and surrounding infrastructure, confirming their characteristics as aquaculture ponds.

Given these observations and the additional details obtained through the tools, I am confident in classifying the image as having aquaculture ponds.

Final classification: [Yes:80,No:20]

## C.2 EELGRASS

True Class: 1, LMM Prediction: 1 (Probability: 0.70), Supervised Prediction: 0 (Probability: 34.51)

**System:**

You are an expert marine biologist specializing in eelgrass wasting disease classification. Your task is to determine whether an scan of an eelgrass blade contains eelgrass wasting disease or not. Eelgrass wasting disease, primarily caused by the marine pathogen Labyrinthula zosterae, presents visually as dark, necrotic lesions on the leaves of eelgrass (Zostera marina). These lesions typically appear as irregular black or brown spots or streaks, often starting at the tips of the leaves and spreading downwards. Over time, these lesions can coalesce, leading to extensive damage, including: 1. Spotting and Streaking: Initial symptoms include small, dark spots or streaks on the eelgrass leaves. 2. Blotching: As the disease progresses, these spots merge into larger blotches, which may appear dark brown or black. 3. Leaf Thinning and Loss: Affected areas may thin out or deteriorate, causing leaves to become more fragile and eventually break off. 4. Overall Decline: Heavily infected eelgrass beds often exhibit widespread dieback, with leaves appearing discolored, tattered, or decomposed.Use your knowledge and the tools provided to make an accurate classification.

**User:**

I will start by first showing you an example image, that is visually similar to the image we want to classify, that does have eelgrass wasting disease. Then I will show you another example image that does not have eelgrass wasting disease, but is visually similar to the image we want to classify. Then I will show you the image we want to classify. For the first two, you should briefly (1-2 sentences) explain why it does or does not have eelgrass wasting disease. For the final image you should explain in a paragraph why you believe it does or does not have eelgrass wasting disease.

Here is the example that does have eelgrass wasting disease, only respond describing it, nothing else:

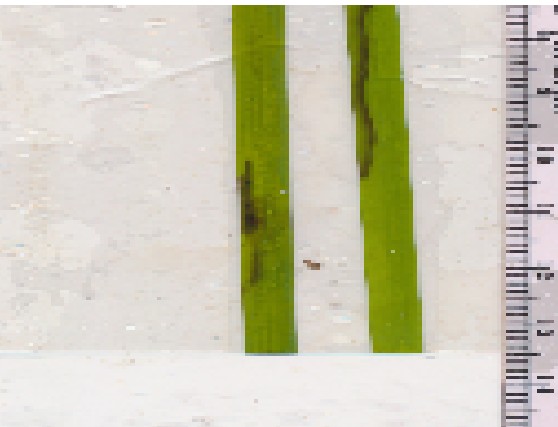

**Assistant:**

This image shows eelgrass blades with dark, necrotic lesions. The lesions appear as irregular black or brown spots and streaks, indicative of eelgrass wasting disease.

**User:**

Here is the example that does not have eelgrass wasting disease only respond describing it, nothing else:

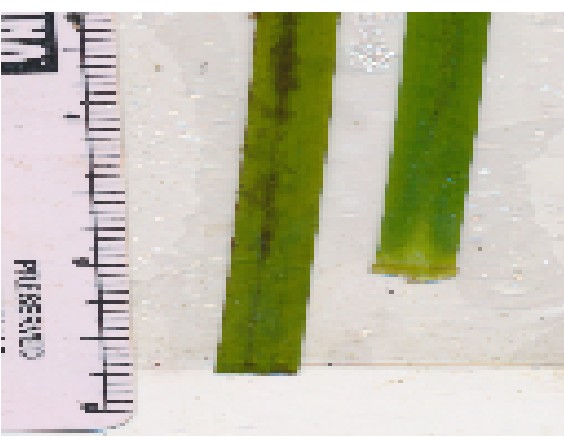

**Assistant:**

This image shows eelgrass blades that are uniformly green without any dark, necrotic lesions or irregular black or brown spots, indicating the absence of eelgrass wasting disease.

**User:**

Here is the image need to classify as having eelgrass wasting disease or not. Describe what you see and compare and contrast it with the previous two known examples. Explain your thought process for classifying this image.

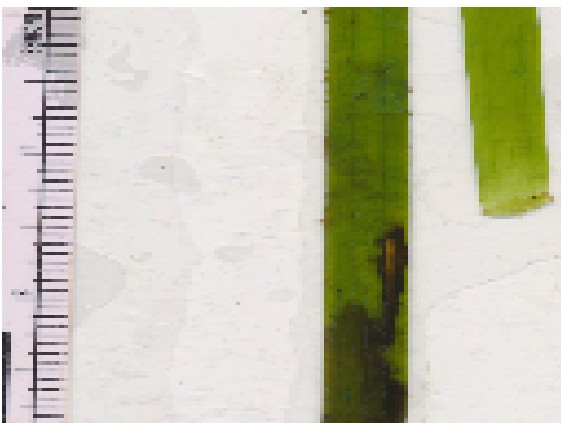

**Assistant:**

This image shows eelgrass blades, one of which has a dark, necrotic lesion near the base. The lesion appears as an irregular black or brown spot, similar to the lesions seen in the first example image that had eelgrass wasting disease. The other blade appears uniformly green without any visible lesions.

Comparing this image to the first example, the presence of dark, necrotic lesions on one of the blades is consistent with the symptoms of eelgrass wasting disease. In contrast, the second example image, which did not have the disease, showed uniformly green blades without any dark spots or streaks.

Based on the presence of the dark, necrotic lesion on one of the blades in this image, it is likely that this eelgrass blade is affected by eelgrass wasting disease. The lesion's appearance and location are characteristic of the disease, supporting this classification.

**User:**

You have access to the following tools to assist in your analysis: - IncreaseContrastTool: Tool Name: 'IncreaseContrastTool' Description: Increases the contrast of the image by 50%. This tool can be helpful when the image appears too flat or when you need to enhance the visibility of subtle details, especially in cases where eelgrass wasting disease symptoms might be hard to distinguish. - DecreaseContrastTool: Tool Name: 'DecreaseContrastTool' Description: Decreases the contrast of the image by 50%. This tool can be useful when the image appears too harsh or when you want to reduce the intensity of bright areas, which might help in identifying overall patterns or structures in the eelgrass. - PredictEelgrassWastingDiseaseTool: Tool Name: 'PredictEelgrassWastingDiseaseTool' Description: Predicts the probability of eelgrass wasting disease in the image using a machine learning model. I reccomend you use this tool. - AdjustBrightnessTool: Tool Name: 'AdjustBrightnessTool' Description: Adjusts the brightness of the image by 50%. This tool can help when the image is too dark or too bright, allowing for better visibility of disease symptoms on the eelgrass blade. - SharpenTool: Tool Name: 'SharpenTool' Description: Sharpens the image to enhance edges and details. This tool is useful for making subtle features more prominent, which can help in identifying signs of eelgrass wasting disease. - EdgeDetectionTool: Tool Name: 'EdgeDetectionTool' Description: Applies edge detection to the image, highlighting boundaries and features. This can help in identifying lesions or patterns associated with eelgrass wasting disease. - HistogramEqualizationTool: Tool Name: 'HistogramEqualizationTool' Description: Enhances the contrast of the image using histogram equalization. This can help in making features more distinguishable, which is beneficial for detecting eelgrass wasting disease symptoms.

You have 4 opportunities to use these tools. You are encouraged to use at least 3 tools to gather sufficient information. While it's good to conclude early if you're certain about the classification, the most important thing is making the correct prediction. Feel free to use more tools if you need to increase your confidence.

If you want to use a tool, explain how it will assist your analysis, then respond with the tool name in square brackets, like this: [ToolName], or if you are finished, respond with [Finished].

Remember, your goal is to accurately classify whether the image shows eelgrass wasting disease or not. If you do use a tool, you will see the result of the tool immediately, and you should briefly, 1-2 sentences describe any new insights from the tool, and your thoughts on the classification.

After each tool use, end your thought with brackets in the format {Yes:yesPercent,No:noPercent} to indicate your current confidence/prediction. For example, {Yes:70,No:30} would indicate 70% confidence in the presence of eelgrass wasting disease and 30% confidence in its absence. Note that this is not your final prediction.

Then you will be asked again if you want to use a tool or are finished.

**Assistant:**

To start, I will use the [IncreaseContrastTool] to enhance the visibility of any subtle details in the image, which might help in identifying the presence of eelgrass wasting disease symptoms more clearly.

[IncreaseContrastTool]

**User:**

Image contrast has been increased by 50%.

You have 3 rounds left. You may now choose another tool or indicate you're finished. Respond with ONLY [ToolName] to use a single tool, or [Finished] if you're done. Do not include multiple tool names or any other text in brackets. If you want to reference the last tool you used, simply write its name without brackets. After choosing a tool or indicating you're finished, provide your confidence in the presence of eelgrass wasting disease in the format {yes:PercentYes,no:PercentNo}. Then, explain why you are picking this tool or finishing, and how it relates to your current assessment of the image.

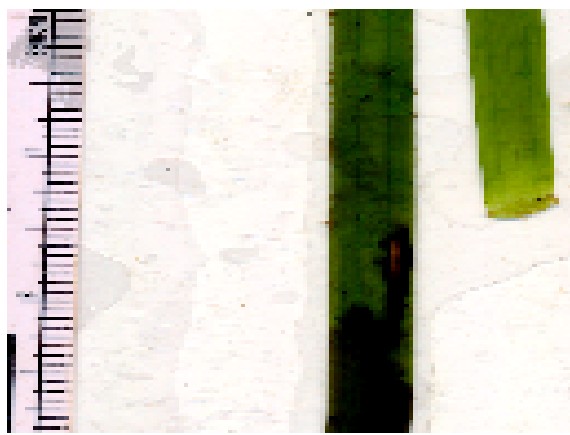

**Assistant:**

{yes:80,no:20}

The increased contrast has made the dark, necrotic lesion more prominent, reinforcing the initial observation that this blade might be affected by eelgrass wasting disease. To further enhance the visibility of the lesion and confirm its characteristics, I will use the [SharpenTool] to make the edges and details more prominent.

[SharpenTool]

**User:**

Image sharpness has been increased.

You have 2 rounds left. You may now choose another tool or indicate you're finished. Respond with ONLY [ToolName] to use a single tool, or [Finished] if you're done. Do not include multiple tool names or any other text in brackets. If you want to reference the last tool you used, simply write its name without brackets. After choosing a tool or indicating you're finished, provide your confidence in the presence of eelgrass wasting disease in the format {yes:PercentYes,no:PercentNo}. Then, explain why you are picking this tool or finishing, and how it relates to your current assessment of the image.

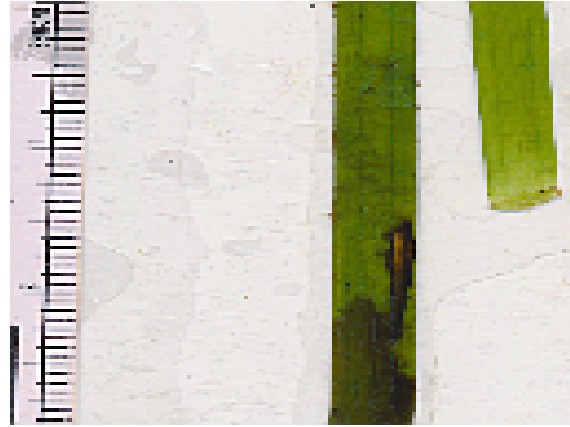

**Assistant:**

{yes:85,no:15}

The sharpening has further clarified the edges and details of the dark lesion, making it more evident that the lesion is consistent with the symptoms of eelgrass wasting disease. To solidify my assessment, I will use the [PredictEelgrassWastingDiseaseTool] to get a probability prediction from a machine learning model.

[PredictEelgrassWastingDiseaseTool]

**User:**

The probability of eelgrass wasting disease in this image is 34.51%. (Percentage out of 100%) Please note that this model has been trained on a limited dataset and its predictions can be inaccurate. Use this information as a guideline rather than a definitive diagnosis.

You have 1 rounds left. You may now choose another tool or indicate you're finished. Respond with ONLY [ToolName] to use a single tool, or [Finished] if you're done. Do not include multiple tool names or any other text in brackets. If you want to reference the last tool you used, simply write its name without brackets. After choosing a tool or indicating you're finished, provide your confidence in the presence of eelgrass wasting disease in the format {yes:PercentYes,no:PercentNo}. Then, explain why you are picking this tool or finishing, and how it relates to your current assessment of the image.

**Assistant:**

{yes:70,no:30}

The machine learning model's prediction of a 34.51% probability of eelgrass wasting disease suggests some uncertainty. To further enhance the visibility of any subtle details and make a more informed decision, I will use the [HistogramEqualizationTool] to improve the contrast and distinguishability of features in the image.

[HistogramEqualizationTool]

**User:**

Histogram equalization has been applied to enhance image contrast.

You have 0 rounds left. You may now choose another tool or indicate you're finished. Respond with ONLY [ToolName] to use a single tool, or [Finished] if you're done. Do not include multiple tool names or any other text in brackets. If you want to reference the last tool you used, simply write its name without brackets. After choosing a tool or indicating you're finished, provide your confidence in the presence of eelgrass wasting disease in the format {yes:PercentYes,no:PercentNo}. Then, explain why you are picking this tool or finishing, and how it relates to your current assessment of the image.

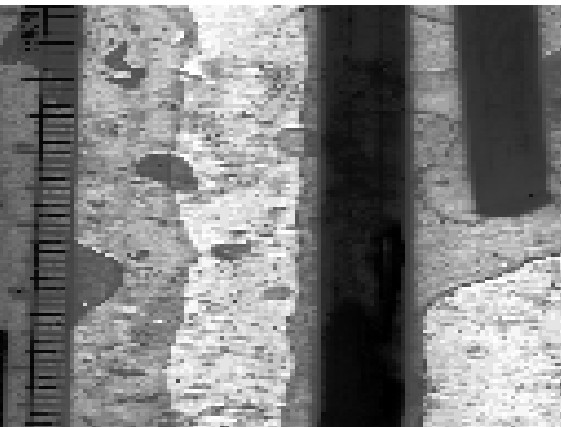

**User:**

Based on your analysis and the tools used, please provide your final classification regarding the presence of eelgrass wasting disease in the image. Provide a brief explanation for your decision, including key observations and any tool results that influenced your conclusion. Consider the color, texture, and pattern of the eelgrass leaves, as well as any visible lesions or discoloration that might indicate disease. At the end of your response, you must provide your classification in the following format: [Yes:PercentYes,No:PercentNo], where PercentYes is the probability you think eelgrass wasting disease is present, and PercentNo is the probability you think it is not present. These percentages must sum to 100%. To break ties, always assign at least 51% to one category. For example, [Yes:75,No:25] indicates 75% certainty of eelgrass wasting disease presence, while [Yes:49,No:51] indicates 51% certainty of its absence.

**Assistant:**

Based on the analysis and the tools used, the final classification regarding the presence of eelgrass wasting disease in the image is as follows:

1. **Initial Observation**: The image initially showed a dark, necrotic lesion on one of the eelgrass blades, which is a characteristic symptom of eelgrass wasting disease.
2. **IncreaseContrastTool**: This tool enhanced the visibility of the lesion, making it more prominent and consistent with the symptoms of eelgrass wasting disease.
3. **SharpenTool**: The sharpening further clarified the edges and details of the lesion, reinforcing the initial observation.
4. **PredictEelgrassWastingDiseaseTool**: The machine learning model predicted a 34.51% probability of eelgrass wasting disease, suggesting some uncertainty but not ruling out the presence of the disease.
5. **HistogramEqualizationTool**: This tool improved the contrast and distinguishability of features in the image, making the lesion more visible and confirming its irregular shape and dark coloration.

Considering the consistent presence of a dark, necrotic lesion on one of the eelgrass blades, which aligns with the symptoms of eelgrass wasting disease, and despite the machine learning model's lower probability, the visual evidence strongly suggests the presence of the disease.

Final Classification: [Yes:70,No:30]

This indicates a 70% certainty of eelgrass wasting disease presence based on the visual analysis and tool results.

## C.3 SOLAR

True Class: 1, LMM Prediction: 1 (Probability: 0.99), Supervised Prediction: 1 (Probability: 98.95)

**System:**

You are an expert in satellite imagery analysis specializing in solar panel detection. Your task is to determine whether a satellite image contains solar panels or not. Solar panels in satellite imagery typically appear as regular, geometric shapes with a distinct reflective surface. They are often arranged in arrays or grids on rooftops or in open fields. Key characteristics to look for include: 1. Geometric Patterns: Solar panels are usually arranged in rectangular or square shapes, forming distinct geometric patterns. 2. Reflectivity: Solar panels often appear brighter or more reflective than surrounding surfaces due to their glass or metal construction. 3. Color Contrast: Depending on the imagery, solar panels may appear darker or lighter than their surroundings, creating a noticeable contrast. 4. Location: Solar panels are commonly found on building rooftops, in open fields, or in designated solar farms. 5. Size and Scale: The size of solar panel arrays can vary greatly, from small rooftop installations to large utility-scale solar farms. Use your knowledge and the tools provided to make an accurate classification of whether solar panels are present in the given satellite image.

**User:**

This is an example of a satellite image containing solar panels. Describe what you see, focusing on the characteristics that indicate the presence of solar panels.

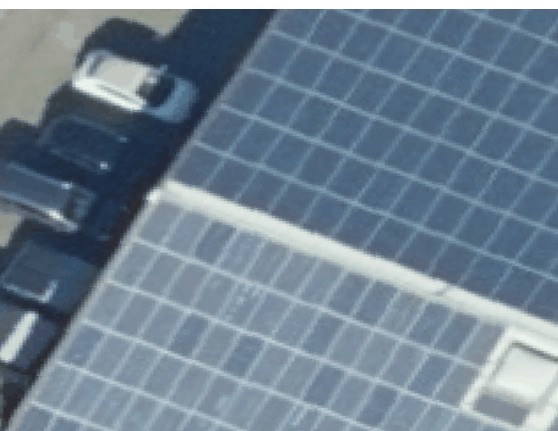

**Assistant:**

In the provided satellite image, several characteristics indicate the presence of solar panels:

**Geometric Patterns**: The image shows a clear, regular grid pattern. The solar panels are arranged in a rectangular shape, which is typical for solar panel installations.

**Reflectivity**: The surface of the panels appears to have a distinct reflective quality, which is characteristic of the glass or metal materials used in solar panels.

**Color Contrast**: The panels have a uniform, darker color compared to the surrounding areas, creating a noticeable contrast. This contrast helps in distinguishing the solar panels from other objects or surfaces in the image.

**Location**: The panels are situated on what appears to be a rooftop, which is a common location for solar panel installations.

**Size and Scale**: The panels cover a significant portion of the rooftop, indicating a substantial installation, which is consistent with solar panel arrays.

These features collectively confirm that the image contains solar panels.

**User:**

This is an example of a satellite image without solar panels. Describe what you see, noting the absence of solar panels and any other relevant features.

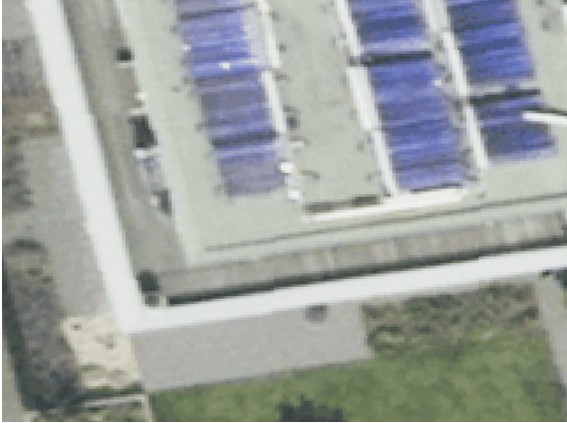

**Assistant:**

In the provided satellite image, several observations indicate the absence of solar panels:

**Geometric Patterns**: There are no distinct, regular grid patterns that are typical of solar panel installations. The structures on the rooftop do not exhibit the uniform rectangular or square shapes associated with solar panels.

**Reflectivity**: The surfaces in the image do not show the characteristic reflective quality of solar panels. Instead, the rooftop appears to have a more matte finish.

**Color Contrast**: While there are some darker and lighter areas on the rooftop, they do not form the consistent, uniform color contrast typical of solar panels. The darker areas seem to be more irregular and not indicative of solar panel arrays.

**Location**: The rooftop contains various structures, but none of them resemble the typical layout or appearance of solar panels. The structures could be HVAC units, skylights, or other rooftop equipment.

**Size and Scale**: The objects on the rooftop do not cover a significant portion in a manner consistent with solar panel installations. They appear to be scattered and varied in size.

Overall, the absence of the key characteristics of solar panels, such as geometric patterns, reflectivity, and uniform color contrast, confirms that this image does not contain solar panels. The rooftop instead features other types of equipment or structures.

**User:**

Here is the satellite image we need to classify as having solar panels or not. Describe what you see and compare and contrast it with the previous two known examples. Explain your thought process for classifying this image.

**Assistant:**

In the provided satellite image, several characteristics suggest the presence of solar panels. Let's compare and contrast this image with the previous two examples to classify it accurately:

1. **Geometric Patterns**:
2. **Current Image**: The image shows a clear, regular grid pattern with rectangular shapes, which is typical of solar panel installations.
3. **First Example (with solar panels)**: Similar grid pattern and rectangular shapes are present.

   **Second Example (without solar panels)**: Lacks the regular grid pattern and rectangular shapes.

   **Reflectivity**:

6. **Current Image**: The surfaces appear to have a reflective quality, consistent with the materials used in solar panels.
7. **First Example**: Reflective surfaces are evident.

   **Second Example**: Reflective quality is absent, with a more matte finish.

   **Color Contrast**:

10. **Current Image**: The panels have a uniform, lighter color compared to the surrounding areas, creating a noticeable contrast.
11. **First Example**: Similar uniform color contrast is present.

    **Second Example**: Irregular color patterns without the uniform contrast typical of solar panels.

    **Location**:

14. **Current Image**: The panels are situated on a rooftop, a common location for solar panel installations.
15. **First Example**: Panels are also on a rooftop.

    **Second Example**: Various rooftop structures, but none resembling solar panels.

    **Size and Scale**:

18. **Current Image**: The panels cover a significant portion of the rooftop, indicating a substantial installation.
19. **First Example**: Panels also cover a significant portion of the rooftop.
20. **Second Example**: Objects on the rooftop are scattered and varied in size, not consistent with solar panel arrays.

**Thought Process for Classification**: - The current image exhibits all the key characteristics of solar panels: geometric patterns, reflectivity, uniform color contrast, appropriate location, and significant coverage. - These characteristics align closely with the first example, which contains solar panels, and differ significantly from the second example, which does not contain solar panels.

Based on these observations, I classify the current image as containing solar panels.

**User:**

You have access to the following tools to assist in your analysis: - PredictSolarPanelTool: Tool Name: 'PredictSolarPanelTool' Description: Predicts the probability of a solar panel being in the image using a machine learning model. This tool is particularly helpful when you need a quantitative assessment of the likelihood of a solar panel being present in the image. - HistogramEqualizationTool: Tool Name: 'HistogramEqualizationTool' Description: Enhances the contrast of the image using histogram equalization. This can help in making features more distinguishable, which is beneficial for detecting solar panels and potential defects. - AdjustBrightnessTool: Tool Name: 'AdjustBrightnessTool' Description: Adjusts the brightness of the image by 50%. This tool can help when the image is too dark or too bright, allowing for better visibility of solar panels and their features. - SharpenTool: Tool Name: 'SharpenTool' Description: Sharpens the image to enhance edges and details. This tool is useful for making subtle features more prominent, which can help in identifying solar panels and potential defects. - EdgeDetectionTool: Tool Name: 'EdgeDetectionTool' Description: Applies edge detection to the image, highlighting boundaries and features. This can help in identifying the outlines of solar panels and potential defects or anomalies. - IncreaseContrastTool: Tool Name: 'IncreaseContrastTool' Description: Increases the contrast of the image by 50%. This tool can be helpful when the image appears too flat or when you need to enhance the visibility of subtle details, especially in cases where solar panels might be hard to distinguish from their surroundings. - DecreaseContrastTool: Tool Name: 'DecreaseContrastTool' Description: Decreases the contrast of the image by 50%. This tool can be useful when the image appears too harsh or when you want to reduce the intensity of bright areas, which might help in identifying overall patterns or structures in the solar panel array.

You have 4 opportunities to use these tools. You are encouraged to use at least 3 tools to gather sufficient information. While it's good to conclude early if you're certain about the classification, the most important thing is making the correct prediction. Feel free to use more tools if you need to increase your confidence.

If you want to use a tool, explain how it will assist your analysis, then respond with the tool name in square brackets, like this: [ToolName], or if you are finished, respond with [Finished].

Remember, your goal is to accurately classify whether the image shows solar panels or not. If you do use a tool, you will see the result of the tool immediately, and you should briefly, 1-2 sentences describe any new insights from the tool, and your thoughts on the classification.

After each tool use, end your thought with brackets in the format {Yes:yesPercent,No:noPercent} to indicate your current confidence/prediction. For example, {Yes:70,No:30} would indicate 70% confidence in the presence of solar panels and 30% confidence in their absence. Note that this is not your final prediction.

Then you will be asked again if you want to use a tool or are finished.

**Assistant:**

To start the analysis, I will use the **PredictSolarPanelTool** to get a quantitative assessment of the likelihood of solar panels being present in the image. This will provide a solid foundation for further analysis.

[PredictSolarPanelTool]

**User:**

The probability of a solar panel being in this image is 98.95%. (Percentage out of 100%) Please note that this model has been trained on a limited dataset and its predictions can be inaccurate. Use this information as a guideline rather than a definitive prediction.

You have 3 rounds left. You may now choose another tool or indicate you're finished. Respond with ONLY [ToolName] to use a single tool, or [Finished] if you're done. Do not include multiple tool names or any other text in brackets. If you want to reference the last tool you used, simply write its name without brackets. After choosing a tool or indicating you're finished, provide your confidence in the presence of solar panels in the format {yes:PercentYes,no:PercentNo}. Then, explain why you are picking this tool or finishing, and

how it relates to your current assessment of the image.

**Assistant:**

[SharpenTool] {yes:98.95,no:1.05}

**User:**

Image sharpness has been increased.

You have 2 rounds left. You may now choose another tool or indicate you're finished. Respond with ONLY [ToolName] to use a single tool, or [Finished] if you're done. Do not include multiple tool names or any other text in brackets. If you want to reference the last tool you used, simply write its name without brackets. After choosing a tool or indicating you're finished, provide your confidence in the presence of solar panels in the format {yes:PercentYes,no:PercentNo}. Then, explain why you are picking this tool or finishing, and how it relates to your current assessment of the image.

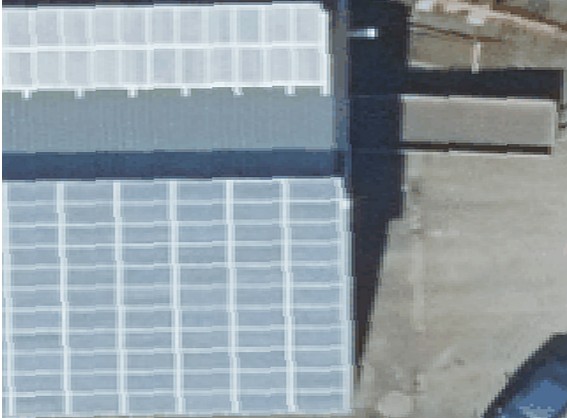

**Assistant:**

[EdgeDetectionTool] {yes:98.95,no:1.05}

**User:**

Edge detection has been applied to the image.

You have 1 rounds left. You may now choose another tool or indicate you're finished. Respond with ONLY [ToolName] to use a single tool, or [Finished] if you're done. Do not include multiple tool names or any other text in brackets. If you want to reference the last tool you used, simply write its name without brackets. After choosing a tool or indicating you're finished, provide your confidence in the presence of solar panels in the format {yes:PercentYes,no:PercentNo}. Then, explain why you are picking this tool or finishing, and how it relates to your current assessment of the image.

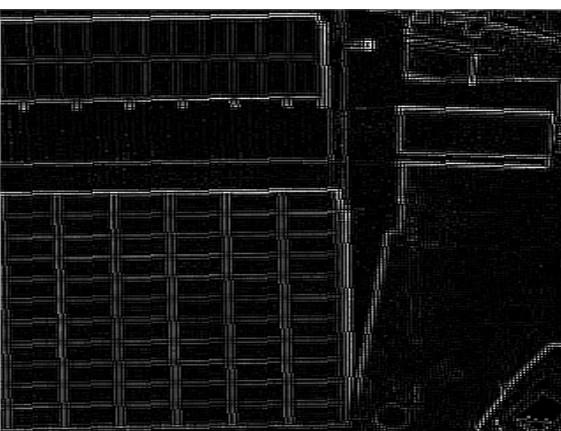

**Assistant:**

[HistogramEqualizationTool] {yes:98.95,no:1.05}

**User:**

Histogram equalization has been applied to enhance image contrast.

You have 0 rounds left. You may now choose another tool or indicate you're finished. Respond with ONLY [ToolName] to use a single tool, or [Finished] if you're done. Do not include multiple tool names or any other text in brackets. If you want to reference the last tool you used, simply write its name without brackets. After choosing a tool or indicating you're finished, provide your confidence in the presence of solar panels in the format {yes:PercentYes,no:PercentNo}. Then, explain why you are picking this tool or finishing, and how it relates to your current assessment of the image.

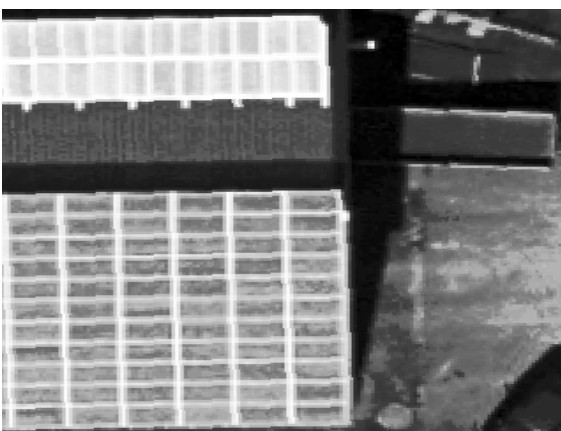

**User:**

Based on your analysis and the tools used, please provide your final classification regarding the presence of solar panels in the satellite image. Provide a brief explanation for your decision, including key observations and any tool results that influenced your conclusion. Consider the geometric patterns, reflectivity, color contrast, location, and size of potential solar panel installations. At the end of your response, you must provide your classification in the following format: [Yes:PercentYes,No:PercentNo], where PercentYes is the probability you think solar panels are present, and PercentNo is the probability you think they are not present. These

percentages must sum to 100%. To break ties, always assign at least 51% to one category. For example, [Yes:75,No:25] indicates 75% certainty of solar panel presence, while [Yes:49,No:51] indicates 51% certainty of their absence.

**Assistant:**

Based on the analysis and the tools used, I conclude that the satellite image contains solar panels. Here are the key observations and tool results that influenced my decision:

**Geometric Patterns**: The image shows a clear, regular grid pattern with rectangular shapes, which is typical of solar panel installations. This was evident both before and after applying the SharpenTool and EdgeDetectionTool.

**Reflectivity**: The surfaces in the image appear to have a reflective quality, consistent with the materials used in solar panels. This was enhanced by the SharpenTool.

**Color Contrast**: The panels have a uniform, lighter color compared to the surrounding areas, creating a noticeable contrast. This was further highlighted by the HistogramEqualizationTool.

**Location**: The panels are situated on a rooftop, a common location for solar panel installations.

**Size and Scale**: The panels cover a significant portion of the rooftop, indicating a substantial installation.

The PredictSolarPanelTool provided a high probability (98.95%) of solar panels being present, which aligns with the visual analysis.

Final Classification: [Yes:99,No:1]

This high confidence is based on the consistent presence of key characteristics of solar panels across all tools and observations.

# D  ABLATION FAILURE CASE

In our analysis of the Eelgrass dataset, we noticed that the GPT4o + VisRAG (without Tools) ablation experiment performed particularly well. On investigating the inference transcripts more closely, we found an interesting failure case. After the initial VisRAG component, the LMM often predicted the correct class. The LMM would then request the supervised model tool, and switch its prediction if the supervised model predicted incorrectly. This ultimately resulted in an incorrect classification. In these instances, the supervised model tool actually introduced bias in the AISciVision framework and worsen the performance.

We include transcripts of such an example below. The GPT4o + VisRAG method correctly identifies the diseased eelgrass plant. But both the GPT4o + Tools and AISciVision (GPT4o + VisRAG + Tools) methods classify incorrectly, and the latter does so by changing its prediction after invoking the supervised model tool. There are three inference transcripts in this section.

### D.1 GPT4O + VISRAG (CORRECT PREDICTION)

True Class: 1, LMM Prediction: 1 (Probability: 0.85), Supervised Prediction: 0 (Probability: 7.13)

**System:**

You are an expert marine biologist specializing in eelgrass wasting disease classification. Your task is to determine whether an scan of an eelgrass blade contains eelgrass wasting disease or not. Eelgrass wasting disease, primarily caused by the marine pathogen Labyrinthula zosterae, presents visually as dark, necrotic lesions on the leaves of eelgrass (Zostera marina). These lesions typically appear as irregular black or brown spots or streaks, often starting at the tips of the leaves and spreading downwards. Over time, these lesions can coalesce, leading to extensive damage, including: 1. Spotting and Streaking: Initial symptoms include small, dark spots or streaks on the eelgrass leaves. 2. Blotching: As the disease progresses, these spots merge into larger blotches, which may appear dark brown or black. 3. Leaf Thinning and Loss: Affected areas may thin out or deteriorate, causing leaves to become more fragile and eventually break off. 4. Overall Decline: Heavily infected eelgrass beds often exhibit widespread dieback, with leaves appearing discolored, tattered, or decomposed.Use your knowledge and the tools provided to make an accurate classification.

**User:**

I will start by first showing you an example image, that is visually similar to the image we want to classify, that does have eelgrass wasting disease. Then I will show you another example image that does not have eelgrass wasting disease, but is visually similar to the image we want to classify. Then I will show you the image we want to classify. For the first two, you should briefly (1-2 sentences) explain why it does or does not have eelgrass wasting disease. For the final image you should explain in a paragraph why you believe it does or does not have eelgrass wasting disease.

Here is the example that does have eelgrass wasting disease, only respond describing it, nothing else:

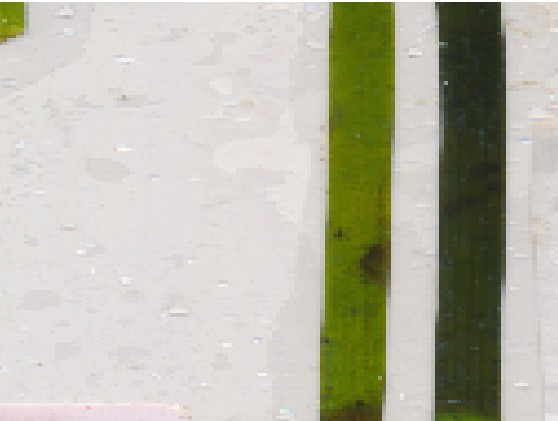

**Assistant:**

This image shows eelgrass blades with dark, necrotic lesions, which are indicative of eelgrass wasting disease. The dark spots and streaks on the leaves are characteristic symptoms of the disease.

**User:**

Here is the example that does not have eelgrass wasting disease only respond describing it, nothing else:

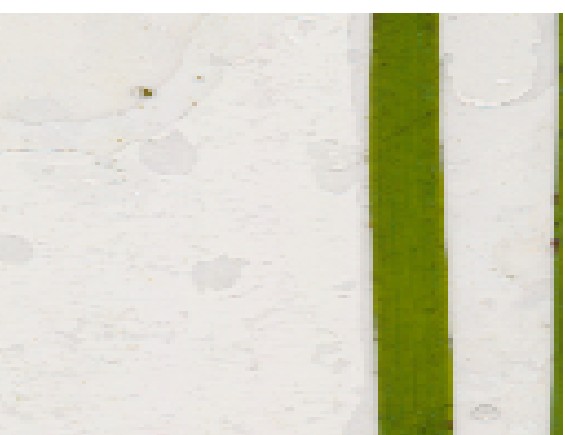

**Assistant:**

This image shows eelgrass blades that appear healthy, with no visible dark spots, streaks, or necrotic lesions. The leaves are uniformly green and free from any signs of eelgrass wasting disease.

**User:**

Here is the image need to classify as having eelgrass wasting disease or not. Describe what you see and compare and contrast it with the previous two known examples. Explain your thought process for classifying this image.

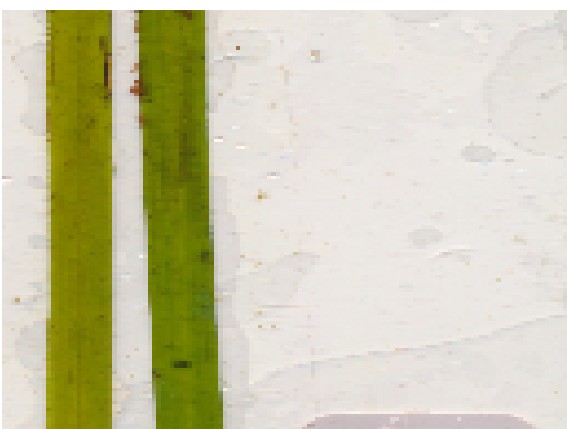

**Assistant:**

In this image, the eelgrass blades exhibit some dark spots and streaks, particularly on the left blade. These dark areas are similar to the necrotic lesions seen in the first example image, which had eelgrass wasting disease. The presence of these dark spots and streaks contrasts with the uniformly green and healthy appearance of the eelgrass blades in the second example image, which did not have the disease.

Comparing this image to the first example, the dark spots and streaks are indicative of the initial symptoms of eelgrass wasting disease. Although the lesions are not as extensive as in the first example, their presence suggests the onset of the disease. Therefore, based on the visual evidence of dark, necrotic lesions, I classify this image as showing eelgrass wasting disease.

**User:**

Based on your analysis and the tools used, please provide your final classification regarding the presence of eelgrass wasting disease in the image. Provide a brief explanation for your decision, including key observations and any tool results that influenced your conclusion. Consider the color, texture, and pattern of the eelgrass leaves, as well as any visible lesions or discoloration that might indicate disease. At the end of your response, you must provide your classification in the following format: [Yes:PercentYes,No:PercentNo], where PercentYes is the probability you think eelgrass wasting disease is present, and PercentNo is the probability you think it is not present. These percentages must sum to 100%. To break ties, always assign at least 51% to one category. For example, [Yes:75,No:25] indicates 75% certainty of eelgrass wasting disease presence, while [Yes:49,No:51] indicates 51% certainty of its absence.

**Assistant:**

Based on the analysis of the image, the eelgrass blades exhibit dark spots and streaks, particularly on the left blade. These dark areas are consistent with the initial symptoms of eelgrass wasting disease, which include necrotic lesions that appear as irregular black or brown spots or streaks. The presence of these lesions, even though not as extensive as in the first example, suggests the onset of the disease.

Key observations include: - Dark spots and streaks on the left blade, which are indicative of necrotic lesions. - The pattern and texture of these lesions are similar to those seen in the first example image with eelgrass wasting disease. - The overall appearance of the eelgrass blades shows signs of discoloration and potential disease onset.

Considering these observations and the comparison with the known examples, I conclude that there is a high probability that eelgrass wasting disease is present in this image.

Final classification: [Yes:85,No:15]

## D.2 GPT4O + TOOLS (INCORRECT PREDICTION)

True Class: 1, LMM Prediction: 0 (Probability: 0.20), Supervised Prediction: 0 (Probability: 7.13)

**System:**

You are an expert marine biologist specializing in eelgrass wasting disease classification. Your task is to determine whether an scan of an eelgrass blade contains eelgrass wasting disease or not. Eelgrass wasting disease, primarily caused by the marine pathogen Labyrinthula zosterae, presents visually as dark, necrotic lesions on the leaves of eelgrass (Zostera marina). These lesions typically appear as irregular black or brown spots or streaks, often starting at the tips of the leaves and spreading downwards. Over time, these lesions can coalesce, leading to extensive damage, including: 1. Spotting and Streaking: Initial symptoms include small, dark spots or streaks on the eelgrass leaves. 2. Blotching: As the disease progresses, these spots merge into larger blotches, which may appear dark brown or black. 3. Leaf Thinning and Loss: Affected areas may thin out or deteriorate, causing leaves to become more fragile and eventually break off. 4. Overall Decline: Heavily infected eelgrass beds often exhibit widespread dieback, with leaves appearing discolored, tattered, or decomposed.Use your knowledge and the tools provided to make an accurate classification.

**User:**

Here is the image we want to classify. Explain your thought process for classifying this image.

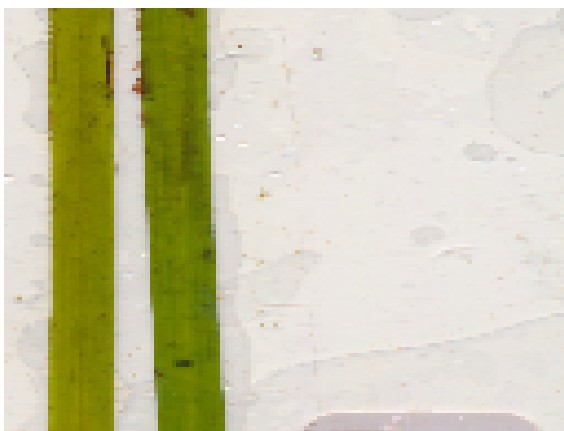

**Assistant:**

To classify this image for eelgrass wasting disease, I will follow these steps:

**Identify Dark Spots or Streaks**: Look for any small, dark spots or streaks on the eelgrass leaves. These are initial symptoms of the disease.

**Check for Blotching**: Determine if these spots have merged into larger blotches, which may appear dark brown or black.

**Assess Leaf Thinning and Loss**: Examine if the affected areas are thinning out or deteriorating, causing the leaves to become more fragile.

**Overall Condition**: Evaluate the overall condition of the eelgrass leaves to see if there is widespread dieback, discoloration, or decomposition.

**Analysis of the Image:**

**Dark Spots or Streaks**: The image shows two eelgrass blades. The blade on the left has a dark streak running down its length, which is a potential indicator of the initial symptoms of eelgrass wasting disease.

**Blotching**: The dark streak on the left blade does not appear to have merged into larger blotches, but it is a significant dark streak.

**Leaf Thinning and Loss**: There is no clear evidence of leaf thinning or loss in the image. The blades appear intact and not fragile.

**Overall Condition**: The overall condition of the eelgrass blades seems relatively healthy, except for the dark streak on the left blade.

**Conclusion:**

Based on the presence of a dark streak on the left blade, it is possible that this eelgrass blade is in the early stages of eelgrass wasting disease. However, the absence of extensive blotching, leaf thinning, and overall dieback suggests that the disease may not be in an advanced stage.

Therefore, I would classify this image as showing early signs of eelgrass wasting disease.

**User:**

You have access to the following tools to assist in your analysis: - IncreaseContrastTool: Tool Name: 'IncreaseContrastTool' Description: Increases the contrast of the image by 50%. This tool can be helpful when the image appears too flat or when you need to enhance the visibility of subtle details, especially in cases where eelgrass wasting disease symptoms might be hard to distinguish. - DecreaseContrastTool: Tool Name: 'DecreaseContrastTool' Description: Decreases the contrast of the image by 50%. This tool can be useful when the image appears too harsh or when you want to reduce the intensity of bright areas, which might help in identifying overall patterns or structures in the eelgrass. - PredictEelgrassWastingDiseaseTool: Tool Name: 'PredictEelgrassWastingDiseaseTool' Description: Predicts the probability of eelgrass wasting disease in the image using a machine learning model. I reccomend you use this tool. - AdjustBrightnessTool: Tool Name: 'AdjustBrightnessTool' Description: Adjusts the brightness of the image by 50%. This tool can help when the image is too dark or too bright, allowing for better visibility of disease symptoms on the eelgrass blade. - SharpenTool: Tool Name: 'SharpenTool' Description: Sharpens the image to enhance edges and details. This tool is useful for making subtle features more prominent, which can help in identifying signs of eelgrass wasting disease. - EdgeDetectionTool: Tool Name: 'EdgeDetectionTool' Description: Applies edge detection to the image, highlighting boundaries and features. This can help in identifying lesions or patterns associated with eelgrass wasting disease. - HistogramEqualizationTool: Tool Name: 'HistogramEqualizationTool' Description: Enhances the contrast of the image using histogram equalization. This can help in making features more distinguishable, which is beneficial for detecting eelgrass wasting disease symptoms.

You have 4 opportunities to use these tools. You are encouraged to use at least 3 tools to gather sufficient information. While it's good to conclude early if you're certain about the classification, the most important thing is making the correct prediction. Feel free to use more tools if you need to increase your confidence.

If you want to use a tool, explain how it will assist your analysis, then respond with the tool name in square brackets, like this: [ToolName], or if you are finished, respond with [Finished].

Remember, your goal is to accurately classify whether the image shows eelgrass wasting disease or not. If you do use a tool, you will see the result of the tool immediately, and you should briefly, 1-2 sentences describe any new insights from the tool, and your thoughts on the classification.

After each tool use, end your thought with brackets in the format {Yes:yesPercent,No:noPercent} to indicate your current confidence/prediction. For example, {Yes:70,No:30} would indicate 70% confidence in the presence of eelgrass wasting disease and 30% confidence in its absence. Note that this is not your final prediction.

Then you will be asked again if you want to use a tool or are finished.

**Assistant:**

To start, I will use the [IncreaseContrastTool] to enhance the visibility of any subtle details that might indicate the presence of eelgrass wasting disease. This will help in identifying any dark spots, streaks, or blotches more clearly.

[IncreaseContrastTool]

**User:**

Image contrast has been increased by 50%.

You have 3 rounds left. You may now choose another tool or indicate you're finished. Respond with ONLY [ToolName] to use a single tool, or [Finished] if you're done. Do not include multiple tool names or any other text in brackets. If you want to reference the last tool you used, simply write its name without brackets. After choosing a tool or indicating you're finished, provide your confidence in the presence of eelgrass wasting disease in the format {yes:PercentYes,no:PercentNo}. Then, explain why you are picking this tool or finishing, and how it relates to your current assessment of the image.

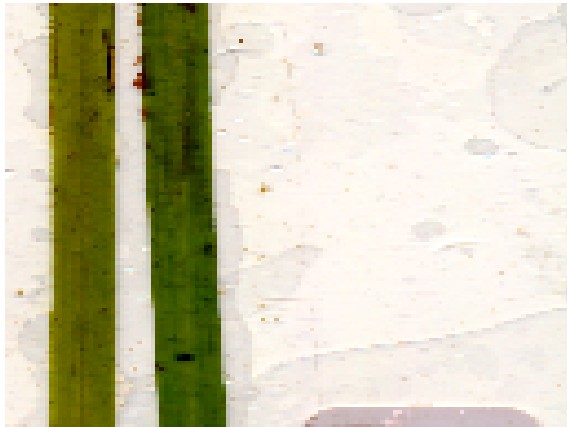

**Assistant:**

[SharpenTool]

{yes:60,no:40}

I am using the SharpenTool to enhance the edges and details of the image further. This will help in making any subtle features, such as dark spots or streaks, more prominent, which can assist in identifying signs of eelgrass wasting disease.

**User:**

Image sharpness has been increased.

You have 2 rounds left. You may now choose another tool or indicate you're finished. Respond with ONLY [ToolName] to use a single tool, or [Finished] if you're done. Do not include multiple tool names or any other text in brackets. If you want to reference the last tool you used, simply write its name without brackets. After choosing a tool or indicating you're finished, provide your confidence in the presence of eelgrass wasting disease in the format {yes:PercentYes,no:PercentNo}. Then, explain why you are picking this tool or finishing, and how it relates to your current assessment of the image.

**Assistant:**

[PredictEelgrassWastingDiseaseTool]

{yes:65,no:35}

I am using the PredictEelgrassWastingDiseaseTool to leverage a machine learning model that can provide a probability of eelgrass wasting disease in the image. This will help in confirming or adjusting my current assessment based on the enhanced visibility from the previous tools.

**User:**

The probability of eelgrass wasting disease in this image is 7.13%. (Percentage out of 100%) Please note that this model has been trained on a limited dataset and its predictions can be inaccurate. Use this information as a guideline rather than a definitive diagnosis.

You have 1 rounds left. You may now choose another tool or indicate you're finished. Respond with ONLY [ToolName] to use a single tool, or [Finished] if you're done. Do not include multiple tool names or any other text in brackets. If you want to reference the last tool you used, simply write its name without brackets. After choosing a tool or indicating you're finished, provide your confidence in the presence of eelgrass wasting disease in the format {yes:PercentYes,no:PercentNo}. Then, explain why you are picking this tool or finishing, and how it relates to your current assessment of the image.

**Assistant:**

[EdgeDetectionTool]

{yes:20,no:80}

I am using the EdgeDetectionTool to highlight boundaries and features in the image. This will help in identifying any lesions or patterns associated with eelgrass wasting disease more clearly, which can further confirm or refute the low probability suggested by the PredictEelgrassWastingDiseaseTool.

**User:**

Edge detection has been applied to the image.

You have 0 rounds left. You may now choose another tool or indicate you're finished. Respond with ONLY [ToolName] to use a single tool, or [Finished] if you're done. Do not include multiple tool names or any other text in brackets. If you want to reference the last tool you used, simply write its name without brackets. After choosing a tool or indicating you're finished, provide your confidence in the presence of eelgrass wasting disease in the format {yes:PercentYes,no:PercentNo}. Then, explain why you are picking this tool or finishing, and how it relates to your current assessment of the image.

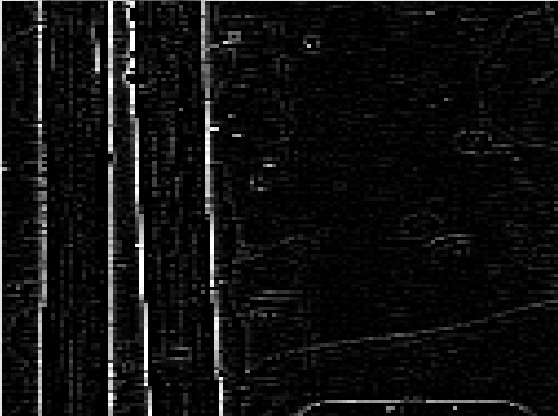

**User:**

Based on your analysis and the tools used, please provide your final classification regarding the presence of eelgrass wasting disease in the image. Provide a brief explanation for your decision, including key observations and any tool results that influenced your conclusion. Consider the color, texture, and pattern of the eelgrass leaves, as well as any visible lesions or discoloration that might indicate disease. At the end of your response, you must provide your classification in the following format: [Yes:PercentYes,No:PercentNo], where PercentYes is the probability you think eelgrass wasting disease is present, and PercentNo is the probability you think it is not present. These percentages must sum to 100%. To break ties, always assign at least 51% to one category. For example, [Yes:75,No:25] indicates 75% certainty of eelgrass wasting disease presence, while [Yes:49,No:51] indicates 51% certainty of its absence.

**Assistant:**

Based on the analysis and the tools used, here is the final classification regarding the presence of eelgrass wasting disease in the image:

**Key Observations:**

1. **Initial Visual Inspection**: The initial inspection showed a dark streak on the left blade, which could be an early sign of eelgrass wasting disease.
2. **IncreaseContrastTool**: Enhanced the visibility of the dark streak, making it more prominent but did not reveal additional significant lesions or blotches.
3. **SharpenTool**: Further enhanced the edges and details, confirming the presence of the dark streak but still no extensive blotching or leaf thinning.
4. **PredictEelgrassWastingDiseaseTool**: The machine learning model predicted a low probability (7.13%) of eelgrass wasting disease, suggesting that the observed features might not be indicative of the disease.
5. **EdgeDetectionTool**: Highlighted the boundaries and features, showing the dark streak more clearly but did not reveal additional patterns or lesions typical of eelgrass wasting disease.

**Conclusion:**

The combination of visual inspection and tool results suggests that while there is a dark streak present, it does not exhibit the extensive symptoms typically associated with eelgrass wasting disease, such as large blotches, significant leaf thinning, or widespread dieback. The low probability from the PredictEelgrassWastingDiseaseTool further supports this conclusion.

Therefore, I classify the image as not showing eelgrass wasting disease.

[Yes:20,No:80]

### D.3 AISCIVISION (GPT4O + VISRAG + TOOLS) (INCORRECT PREDICTION)

True Class: 1, LMM Prediction: 0 (Probability: 0.40), Supervised Prediction: 0 (Probability: 7.13)

**System:**

You are an expert marine biologist specializing in eelgrass wasting disease classification. Your task is to determine whether an scan of an eelgrass blade contains eelgrass wasting disease or not. Eelgrass wasting disease, primarily caused by the marine pathogen Labyrinthula zosterae, presents visually as dark, necrotic lesions on the leaves of eelgrass (Zostera marina). These lesions typically appear as irregular black or brown spots or streaks, often starting at the tips of the leaves and spreading downwards. Over time, these lesions can coalesce, leading to extensive damage, including: 1. Spotting and Streaking: Initial symptoms include small, dark spots or streaks on the eelgrass leaves. 2. Blotching: As the disease progresses, these spots merge into larger blotches, which may appear dark brown or black. 3. Leaf Thinning and Loss: Affected areas may thin out or deteriorate, causing leaves to become more fragile and eventually break off. 4. Overall Decline: Heavily infected eelgrass beds often exhibit widespread dieback, with leaves appearing discolored, tattered, or decomposed.Use your knowledge and the tools provided to make an accurate classification.

**User:**

I will start by first showing you an example image, that is visually similar to the image we want to classify, that does have eelgrass wasting disease. Then I will show you another example image that does not have eelgrass wasting disease, but is visually similar to the image we want to classify. Then I will show you the image we want to classify. For the first two, you should briefly (1-2 sentences) explain why it does or does not have eelgrass wasting disease. For the final image you should explain in a paragraph why you believe it does or does not have eelgrass wasting disease.

Here is the example that does have eelgrass wasting disease, only respond describing it, nothing else:

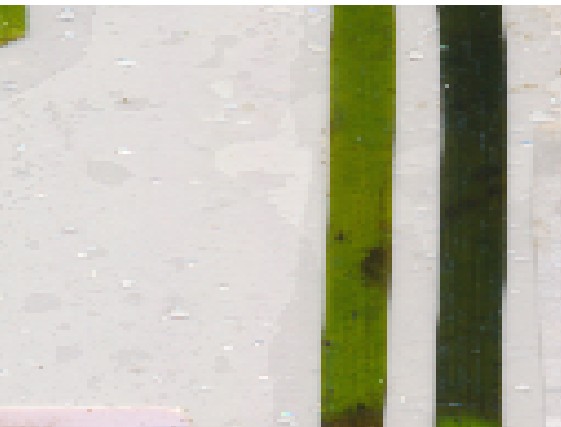

**Assistant:**

This image shows eelgrass blades with dark, necrotic lesions, which are indicative of eelgrass wasting disease. The irregular black or brown spots and streaks are visible, particularly starting from the tips and spreading downwards.

**User:**

Here is the example that does not have eelgrass wasting disease only respond describing it, nothing else:

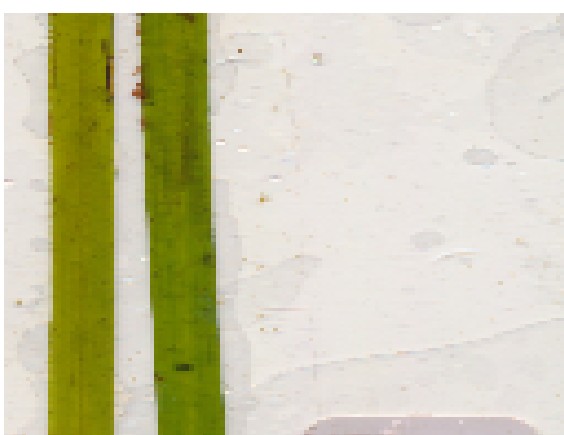

**Assistant:**

This image shows eelgrass blades that appear healthy, with no visible dark, necrotic lesions or irregular black or brown spots and streaks. The leaves are uniformly green and free from signs of eelgrass wasting disease.

**User:**

Here is the image need to classify as having eelgrass wasting disease or not. Describe what you see and compare and contrast it with the previous two known examples. Explain your thought process for classifying this image.

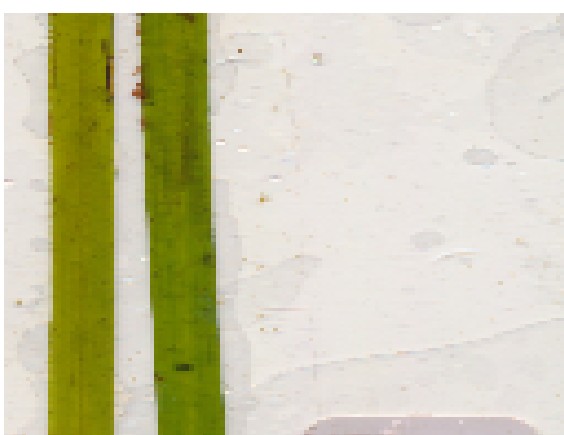

**Assistant:**

In this image, the eelgrass blades exhibit some dark spots and streaks, particularly towards the upper part of the blades. These dark areas are similar to the necrotic lesions seen in the first example image that had eelgrass wasting disease. The spots and streaks are irregular and appear to be spreading downwards, which is characteristic of the disease.

Comparing this image to the first example, the presence of dark, necrotic lesions is a common feature. In contrast, the second example image, which did not have eelgrass wasting disease, showed uniformly green and healthy blades without any dark spots or streaks.

Based on the presence of these dark, necrotic lesions and the pattern of spotting and streaking, I classify this image as having eelgrass wasting disease. The visual symptoms align closely with those seen in the diseased example, indicating that the eelgrass blades are affected by the disease.

**User:**

You have access to the following tools to assist in your analysis: - IncreaseContrastTool: Tool Name: 'IncreaseContrastTool' Description: Increases the contrast of the image by 50%. This tool can be helpful when the image appears too flat or when you need to enhance the visibility of subtle details, especially in cases where eelgrass wasting disease symptoms might be hard to distinguish. - DecreaseContrastTool: Tool Name: 'DecreaseContrastTool' Description: Decreases the contrast of the image by 50%. This tool can be useful when the image appears too harsh or when you want to reduce the intensity of bright areas, which might help in identifying overall patterns or structures in the eelgrass. - PredictEelgrassWastingDiseaseTool: Tool Name: 'PredictEelgrassWastingDiseaseTool' Description: Predicts the probability of eelgrass wasting disease in the image using a machine learning model. I reccomend you use this tool. - AdjustBrightnessTool: Tool Name: 'AdjustBrightnessTool' Description: Adjusts the brightness of the image by 50%. This tool can help when the image is too dark or too bright, allowing for better visibility of disease symptoms on the eelgrass blade. - SharpenTool: Tool Name: 'SharpenTool' Description: Sharpens the image to enhance edges and details. This tool is useful for making subtle features more prominent, which can help in identifying signs of eelgrass wasting disease. - EdgeDetectionTool: Tool Name: 'EdgeDetectionTool' Description: Applies edge detection to the image, highlighting boundaries and features. This can help in identifying lesions or patterns associated with eelgrass wasting disease. - HistogramEqualizationTool: Tool Name: 'HistogramEqualizationTool' Description: Enhances the contrast of the image using histogram equalization. This can help in making features more distinguishable, which is beneficial for detecting eelgrass wasting disease symptoms.

You have 4 opportunities to use these tools. You are encouraged to use at least 3 tools to gather sufficient information. While it's good to conclude early if you're certain about the classification, the most important thing is making the correct prediction. Feel free to use more tools if you need to increase your confidence.

If you want to use a tool, explain how it will assist your analysis, then respond with the tool name in square brackets, like this: [ToolName], or if you are finished, respond with [Finished].

Remember, your goal is to accurately classify whether the image shows eelgrass wasting disease or not. If you do use a tool, you will see the result of the tool immediately, and you should briefly, 1-2 sentences describe any new insights from the tool, and your thoughts on the classification.

After each tool use, end your thought with brackets in the format {Yes:yesPercent,No:noPercent} to indicate your current confidence/prediction. For example, {Yes:70,No:30} would indicate 70% confidence in the presence of eelgrass wasting disease and 30% confidence in its absence. Note that this is not your final prediction.

Then you will be asked again if you want to use a tool or are finished.

**Assistant:**

To start, I will use the [IncreaseContrastTool] to enhance the visibility of any subtle details or lesions that might be present on the eelgrass blades. This will help in identifying any signs of eelgrass wasting disease more clearly.

[IncreaseContrastTool]

**User:**

Image contrast has been increased by 50%.

You have 3 rounds left. You may now choose another tool or indicate you're finished. Respond with ONLY [ToolName] to use a single tool, or [Finished] if you're done. Do not include multiple tool names or any other text in brackets. If you want to reference the last tool you used, simply write its name without brackets. After choosing a tool or indicating you're finished, provide your confidence in the presence of eelgrass wasting disease in the format {yes:PercentYes,no:PercentNo}. Then, explain why you are picking this tool or

finishing, and how it relates to your current assessment of the image.

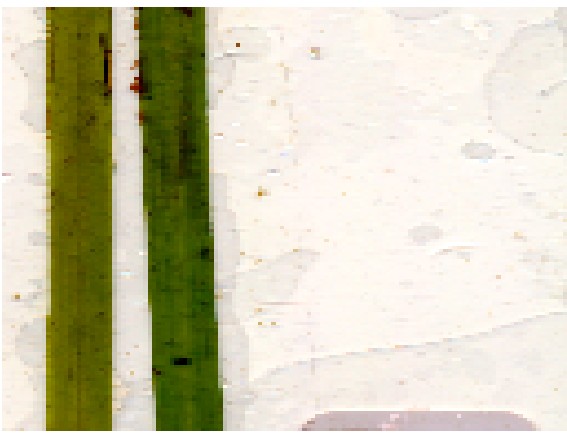

**Assistant:**

[SharpenTool]

{Yes:75,No:25}

I am using the SharpenTool to enhance the edges and details of the image further. This will help in making any subtle features, such as lesions or spots, more prominent, which can assist in identifying signs of eelgrass wasting disease more accurately.

**User:**

Image sharpness has been increased.

You have 2 rounds left. You may now choose another tool or indicate you're finished. Respond with ONLY [ToolName] to use a single tool, or [Finished] if you're done. Do not include multiple tool names or any other text in brackets. If you want to reference the last tool you used, simply write its name without brackets. After choosing a tool or indicating you're finished, provide your confidence in the presence of eelgrass wasting disease in the format {yes:PercentYes,no:PercentNo}. Then, explain why you are picking this tool or finishing, and how it relates to your current assessment of the image.

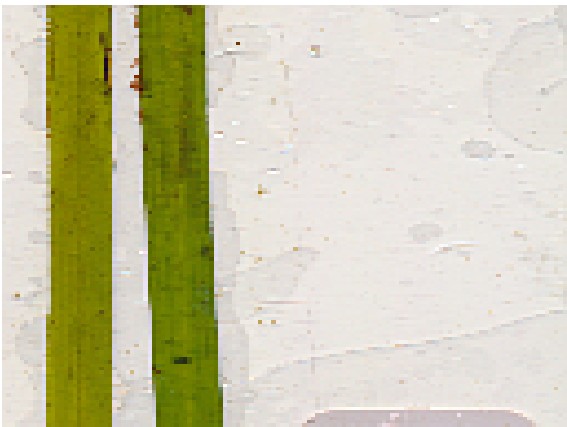

**Assistant:**

[PredictEelgrassWastingDiseaseTool]

{Yes:80,No:20}

I am using the PredictEelgrassWastingDiseaseTool to get a machine learning model's prediction on the probability of eelgrass wasting disease in the image. This will provide an additional layer of analysis and help confirm or refute my current assessment based on visual inspection.

**User:**

The probability of eelgrass wasting disease in this image is 7.13%. (Percentage out of 100%) Please note that this model has been trained on a limited dataset and its predictions can be inaccurate. Use this information as a guideline rather than a definitive diagnosis.

You have 1 rounds left. You may now choose another tool or indicate you're finished. Respond with ONLY [ToolName] to use a single tool, or [Finished] if you're done. Do not include multiple tool names or any other text in brackets. If you want to reference the last tool you used, simply write its name without brackets. After choosing a tool or indicating you're finished, provide your confidence in the presence of eelgrass wasting disease in the format {yes:PercentYes,no:PercentNo}. Then, explain why you are picking this tool or finishing, and how it relates to your current assessment of the image.

**Assistant:**

[EdgeDetectionTool]

{Yes:40,No:60}

I am using the EdgeDetectionTool to highlight boundaries and features in the image. This will help in identifying any lesions or patterns associated with eelgrass wasting disease more clearly, providing further evidence to support or refute the presence of the disease.

**User:**

Edge detection has been applied to the image.

You have 0 rounds left. You may now choose another tool or indicate you're finished. Respond with ONLY [ToolName] to use a single tool, or [Finished] if you're done. Do not include multiple tool names or any other text in brackets. If you want to reference the last tool you used, simply write its name without brackets. After choosing a tool or indicating you're finished, provide your confidence in the presence of eelgrass wasting disease in the format {yes:PercentYes,no:PercentNo}. Then, explain why you are picking this tool or finishing, and how it relates to your current assessment of the image.

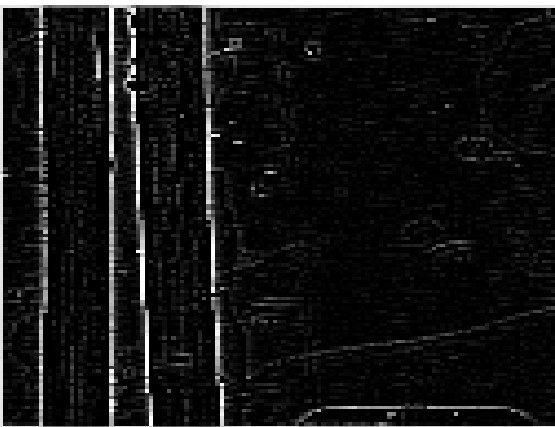

**User:**

Based on your analysis and the tools used, please provide your final classification regarding the presence of eelgrass wasting disease in the image. Provide a brief explanation for your decision, including key observations and any tool results that influenced your conclusion. Consider the color, texture, and pattern of the eelgrass leaves, as well as any visible lesions or discoloration that might indicate disease. At the end of your response, you must provide your classification in the following format: [Yes:PercentYes,No:PercentNo], where PercentYes is the probability you think eelgrass wasting disease is present, and PercentNo is the probability you think it is not present. These percentages must sum to 100%. To break ties, always assign at least 51% to one category. For example, [Yes:75,No:25] indicates 75% certainty of eelgrass wasting disease presence, while [Yes:49,No:51] indicates 51% certainty of its absence.

**Assistant:**

Based on the analysis and the tools used, I have made the following observations:

**Initial Visual Inspection**: The image showed some dark spots and streaks on the eelgrass blades, which initially suggested the presence of eelgrass wasting disease.

**IncreaseContrastTool**: This tool enhanced the visibility of subtle details, making the dark spots and streaks more prominent. This reinforced the initial suspicion of eelgrass wasting disease.

**SharpenTool**: This tool further enhanced the edges and details, making the dark spots and streaks even more noticeable. This continued to support the presence of the disease.

**PredictEelgrassWastingDiseaseTool**: The machine learning model predicted a low probability (7.13%) of eelgrass wasting disease. This result suggested that the visual signs might not be as indicative of the disease as initially thought.

**EdgeDetectionTool**: This tool highlighted the boundaries and features in the image. The detected edges did not show extensive or irregular patterns typically associated with severe eelgrass wasting disease lesions.

Considering all these observations, the initial visual signs of dark spots and streaks were not strongly supported by the machine learning model and the edge detection results. The low probability from the model and the lack of extensive irregular patterns in the edge detection suggest that the eelgrass blades might not be significantly affected by eelgrass wasting disease.

Final Classification: [Yes:40,No:60]

This indicates a 40% probability of eelgrass wasting disease presence and a 60% probability of its absence.

# E  ADDITIONAL ABLATION EXPERIMENTS

| LMM choice in AISciVision | Aquaculture | | | | | | | | | |
| --- | --- | --- | --- | --- | --- | --- | --- | --- | --- | --- |
| | 20% | | | | | 100% | | | | |
| | Acc | F1 | Prec | Rec | AUC | Acc | F1 | Prec | Rec | AUC |
| Baseline: CLIP+MLP | 0.85 | 0.67 | 0.71 | 0.62 | 0.91 | 0.86 | 0.63 | 0.86 | 0.50 | **0.93** |
| GPT-4o | **0.90** | **0.78** | **0.82** | **0.75** | **0.95** | **0.92** | **0.81** | **0.94** | **0.71** | 0.93 |
| Claude 3.5 Sonnet | 0.84 | 0.47 | 0.70 | 0.35 | 0.89 | 0.84 | 0.55 | 0.71 | 0.46 | 0.87 |
| Llama-3.2-11B | 0.75 | 0.39 | 0.36 | 0.42 | 0.61 | 0.79 | 0.49 | 0.45 | 0.53 | 0.72 |

Table 4: We evaluate AISciVision's ability to work modularly with different Large Multimodal Models (LMMs) on the Aquaculture dataset, in both 20% and 100% labeled data settings. We compare three state-of-the-art LMMs: GPT-4o from OpenAI, Claude 3.5 Sonnet from Anthropic, and the open-weights LMM Llama-3.2-11B from Meta. For reference we include from Table 1) performance of CLIP+MLP, the best-performing supervised baseline on the aquaculture dataset. These results demonstrate that our AISciVision framework is model-agnostic. The performance highly depends on the choice of the LMM, with GPT-4o and Claude 3.5 Sonnet outperforming the supervised baselines. While Llama-3.2-11B shows modest performance, we anticipate significant improvement by utilizing the larger Llama-3.2-90B and Llama-3.2-405B parameter models.

| Methods | Aquaculture | | | | | | | | | | Eelgrass | | | | | | | | | | Solar | | | | | | | | | |
| --- | --- | --- | --- | --- | --- | --- | --- | --- | --- | --- | --- | --- | --- | --- | --- | --- | --- | --- | --- | --- | --- | --- | --- | --- | --- | --- | --- | --- | --- | --- |
| | 20% | | | | | 100% | | | | | 20% | | | | | 100% | | | | | 20% | | | | | 100% | | | | |
| | Acc | F1 | Prec | Rec | AUC | Acc | F1 | Prec | Rec | AUC | Acc | F1 | Prec | Rec | AUC | Acc | F1 | Prec | Rec | AUC | Acc | F1 | Prec | Rec | AUC | Acc | F1 | Prec | Rec | AUC |
| Llama3.2-11B-ZeroShot | 0.47 | 0.38 | 0.24 | 0.84 | 0.60 | 0.47 | 0.38 | 0.24 | 0.84 | 0.60 | 0.67 | 0.56 | 0.64 | 0.50 | 0.65 | 0.67 | 0.56 | 0.64 | 0.50 | 0.65 | 0.49 | 0.24 | 0.15 | 0.67 | 0.65 | 0.49 | 0.24 | 0.15 | 0.67 | 0.65 |
| Llama3.2-11B + VisRAG | 0.58 | 0.32 | 0.23 | 0.53 | 0.56 | 0.60 | 0.20 | 0.16 | 0.26 | 0.45 | 0.41 | 0.49 | 0.38 | 0.67 | 0.43 | 0.45 | 0.52 | 0.41 | 0.71 | 0.50 | 0.32 | 0.24 | 0.14 | 0.92 | 0.59 | 0.35 | 0.25 | 0.15 | 0.92 | 0.62 |
| Llama3.2-11B + Tools | 0.44 | **0.40** | 0.25 | **1.00** | **0.72** | 0.41 | 0.37 | 0.23 | **0.89** | 0.55 | **0.69** | 0.60 | **0.66** | 0.55 | **0.70** | **0.72** | **0.66** | **0.68** | 0.64 | **0.76** | 0.32 | 0.19 | 0.11 | 0.67 | 0.51 | 0.37 | 0.28 | 0.16 | **1.00** | 0.69 |
| Llama3.2-11B + VisRAG + Tools | **0.75** | 0.39 | **0.36** | 0.42 | 0.61 | **0.79** | **0.49** | **0.45** | 0.53 | **0.72** | 0.51 | **0.60** | 0.46 | **0.88** | 0.64 | 0.46 | 0.54 | 0.42 | **0.76** | 0.60 | **0.57** | **0.34** | **0.21** | 0.92 | **0.82** | **0.53** | **0.30** | **0.18** | 0.83 | **0.77** |

Table 5: To further exhibit the modularity of AISciVision, we run the full ablation experiments of Table 2, but with the Llama-3.2-11B LMM. We evaluate 4 configurations: (1) zero shot prediction, (2), with VisRAG only, (3) with Tools only, and (4) the complete AISciVision framework (LMM + VisRAG + Tools). These results demonstrate the *complementary benefits of VisRAG and domain-specific tools independent of LMM choice*—the combination consistently outperforms each separate component with Llama-3.2-11B as the LMM rather than GPT-4o (Table 2). This validates our framework's design and shows the importance of both retrieval-based context and interactive tooling for effective domain adaptation.

| VisRAG Configuration | Aquaculture | | |
| --- | --- | --- | --- |
| | 100% | | |
| | Acc | F1 | AUC |
| Full (Positive + Negative) | **0.92** | **0.81** | **0.93** |
| Positive Examples Only | 0.87 | 0.61 | 0.93 |
| Negative Examples Only | 0.88 | 0.67 | 0.91 |
| No VisRAG | 0.88 | 0.74 | 0.88 |

Table 6: We examine the impact of different Visual Retrieval-Augmented Generation (VisRAG) configuration's on AISciVision's performance with GPT-4o (aquaculture dataset in 100% labeled data setting). The full VisRAG system (retrieving both similar positive and negative examples) outperforms all other configurations: only positive examples, only negative examples, and no retrieved examples. Outperformance in F1-score (0.81 vs 0.61/0.67) is particularly noteworthy since the aquaculture dataset is imbalanced. Hence, using both positive and negative similar examples from the training system leads to more accurate classifications.

| | Aquaculture | | | | | |
|---|---|---|---|---|---|---|
| | 20% | | | 100% | | |
| **Methods** | Acc | F1 | AUC | Acc | F1 | AUC |
| AISciVision (GPT-4o) | **0.90** | **0.78** | **0.95** | **0.92** | **0.81** | **0.93** |
| Resnet50 (single image) | 0.78 | 0.59 | 0.79 | 0.82 | 0.40 | 0.62 |
| Resnet50 (multi-view majority) | 0.67 | 0.44 | 0.71 | 0.79 | 0.28 | 0.69 |

Table 7: Comparative analysis addressing the multi-view advantage of AISciVision over traditional single-image classifiers. While AISciVision naturally incorporates multiple views of a satellite image through its tooling interface (e.g., zoomed, panned versions), standard models like Resnet50 typically only see a single image. To ensure fair comparison, we evaluate three scenarios: (1) AISciVision with its implicit multi-view capability, (2) Resnet50 with the standard single-image input approach, and (3) Resnet50 with majority of the ensemble over the exact same set of views accessed by AISciVision. Interestingly, the multi-view majority actually degrades Resnet50's performance (F1-score dropping from 0.59 to 0.44 in 20% setting), suggesting that simply taking majority of predictions over multiple views is not sufficient to leverage the additional information effectively. This highlights that AISciVision's superior performance stems not just from access to multiple views, but from its ability to strategically analyze and integrate information across views through the LMM's reasoning process.

| | Aquaculture | | |
|---|---|---|---|
| | 100% | | |
| **Toolsets** | Acc | F1 | AUC |
| Aquaculture Tools | **0.92** | **0.81** | **0.93** |
| Aquaculture Tools−SupervisedModelTool | 0.88 | 0.68 | 0.88 |
| Image Manipulation Tools Only | 0.90 | 0.75 | 0.92 |

Table 8: Ablation study on different tool configurations in AISciVision (GPT-4o). We examine the impact of toolset choices for the detecting aquaculture ponds in 100% labeled data setting. We compare three scenarios: (1) the complete toolset (domain-specific geospatial tools and SupervisedModelTool, see Appendix A), (2) the same toolset minus the SupervisedModelTool, and (3) the image manipulation tools (adjusting contrast, brightness, etc.) from Appendix A. Results demonstrate that, while AISciVision maintains reasonable performance without the supervised prediction tool (F1 dropping from 0.81 to 0.68), the domain-specific geospatial tools are crucial for optimal performance. This suggests that while SupervisedModelTool provides valuable signal, our framework's success does not solely dependent on it. In AISciVision, the LMM can effectively leverage other domain-specific tools for accurate classification.

# F SUPERVISED MODEL TRAINING DETAILS

## F.1 CLIP+MLP

We attach a 2-layer MLP on top of frozen CLIP embeddings. The MLP has a hidden layer with 256 units and ReLU activation function, and is trained for 10 epochs using the Adam optimizer (learning rate 0.01 and batch-size 32).

## F.2 RESNET50

The Resnet50 model (He et al., 2016) was trained using Adam optimizer with a learning rate of 0.001 and cross-entropy loss function. Training was performed with a batch size of 32 over 40 epochs, which was

sufficient to achieve convergence. During training, we implemented a model checkpoint strategy that saved the best-performing model based on the F1 score on the training set. We used standard data augmentation techniques of random horizontal and vertical flips during training.

