# OpenReview forum: "AISciVision: A Framework for Specializing Large Multimodal Models in Scientific Image Classification"
_ICLR.cc/2025/Conference — Submitted to ICLR 2025_

### Official Review · Reviewer_cXN6 · 2024-10-19

**Soundness:** 2
**Presentation:** 4
**Contribution:** 2
**Rating:** 3
**Confidence:** 5

**Summary:**

This paper proposes a method that uses an LMM, such as ChatGPT, in combination with VisRAG and domain-specific tools to classify scientific images. At test time, given an image, the system retrieves similar images and prompts the LMM to utilize the tools (explaining how to use each one) to better understand and then classify the image. In addition to the classification result, the method also provides the confidence score and the conversation with the LMM to ensure transparency for the user.

**Strengths:**

- The authors try to increase the interpretability of classification task
- Make use of LMMs to solve real world problems
- Does not require any additional model training

**Weaknesses:**

- Comparing new data to similar data carries a high risk of introducing bias into the process.
- Defining domain-specific tools is challenging, as it requires domain experts, and even experts may overlook some tools.
- Using multiple rounds of prediction increases both inference time and cost.
- Describing domain-specific tools in natural language is not always straightforward.
- The testing datasets are very small and may not cover a broad range of test samples.

**Questions:**

- How do you use a hardcoded prompt template for the response?
- How does your model output a probability score indicating its confidence?
- How do you determine how many conversation turns are enough?
- In the paper, you state: "We provide text descriptions of the available tools as prompts to the LMM, and instruct how to request tool use" How do you ensure that the LMM reliably follows your instructions behind the scenes? We know LMMs can sometimes misunderstand prompts, requiring users to read the answer, refine their prompt, and ask again.
- In the paper, you mention: "You ask the LMM to have 4 conversation turns." How do you determine that this number (4) is appropriate?
- In the paper, you state: "All embeddings for VisRAG are computed via CLIP." How do you ensure that these images are not out of distribution for CLIP?
- In the paper, you claim that "AISciVision consistently outperforms all baselines on all three datasets." What about the descriptive text your framework returns? Did you conduct any analysis on the quality of those texts?

---

> ### Author Response · Authors · 2024-11-20
> **Review 4**
>
> Thanks for your candid feedback and suggestions for improvement - we address individual points below:
>
> **Bias**
>
> While we agree and see your point - that comparing inference data to training data introduces bias to the network - we would argue this is exactly the type of bias we do want to encourage in the LLM - to be more accurate/adapt to the dataset at hand. Bias in general plays an interesting role in machine learning, but we fail to see how this issue is specific to our method.
>
> **Domain Specific Tools**
>
> You are right in that this is the most important part to get right when working on real world problems - and we are extremely fortunate that our collaborators are domain experts who helped carefully craft the domain tools, and their natural language description to the model. But you are right to be careful about this - we found this to be one of the most critical parts of getting our work to be truly useful as a deployed application! We hope that while these tools were developed for our domains in particular, they can be readily adapted to many of the other domains that require dealing with satellite imagery, or scientific images.
>
> **Prediction Cost**
>
> Yes this is definitely a trade-off of our method, and improving inference speed and cost is an interesting area of LMM research. However, our domain expert collaborators feel the trade-off is worth it given the higher interpretability.
>
> **Questions:**
>
> 1. *“How do you use a hardcoded prompt template for the response?”*
> Sorry, we are not sure exactly what you mean here - the prompts are crafted to be domain specific for each dataset, and the tooling/visrag prompts are hard coded - but the inputs/outputs are dynamic.
>
> 2. *“How does your model output a probability score indicating its confidence?”*
> We ask the LLM to output an estimate of its probability as a token. This was a design choice on our end and extracting probability estimates from language model outputs is an active area of research.
>
> 3. *“How do you determine how many conversation turns are enough?”*
> At a high-level, we ask the LLM to do this, by allowing at minimum one turn, maximum four. However in practice, we do heavily encourage three turns - this ultimately comes from working with these datasets, representing a tuneable hyperparameter.
>
> 4. *“In the paper, you state: "We provide text descriptions of the available tools as prompts to the LMM, and instruct how to request tool use" How do you ensure that the LMM reliably follows your instructions behind the scenes? We know LMMs can sometimes misunderstand prompts, requiring users to read the answer, refine their prompt, and ask again.”*
> We were concerned with this when originally starting on this work - because should the LLM not follow the input/output patterns we set it would actually break the code as the output would not parse - luckily the LLMs seldom ran into this issue.
>
> 5. *“In the paper, you mention: "You ask the LMM to have 4 conversation turns." How do you determine that this number (4) is appropriate?”*
> Similar to (3) above - this is a hyperparameter decided through experimentation as having a good trade-off. The model can manipulate the data without overloading the framework.
>
> 6. *“In the paper, you state: "All embeddings for VisRAG are computed via CLIP." How do you ensure that these images are not out of distribution for CLIP?”*
> While CLIP is a substantial foundational model trained on many types of imagery, this is definitely a concern, as out-of-distribution data could limit the efficacy of CLIP to embed them in a meaningful way. However, the success of the kNN model suggests that CLIP embeddings are effective in the datasets we study. This is an excellent point, nonetheless, and we are exploring using some satellite-image-specific embedding models for our future work.
>
>
> 7. *“In the paper, you claim that "AISciVision consistently outperforms all baselines on all three datasets." What about the descriptive text your framework returns? Did you conduct any analysis on the quality of those texts?”*
> We do qualitatively get feedback from our domain coauthors about the efficacy and usefulness of the transcript response. Unfortunately, we are still working on a quantitative study based on human interaction, which will take a considerable amount of time.

---

> > ### Author Response · Authors · 2024-11-25
> >
> > Dear Reviewer,
> >
> > Thank you for your detailed analysis. We have thoroughly addressed the questions you raised about prompt reliability, confidence scoring, and conversation dynamics, including empirical validation. We've also (we hope) clarified the questions regarding CLIP embedding distribution. We would greatly value your thoughts on these new results.
> > If these additional experiments and clarifications have addressed your concerns we hope you might consider raising your score. Thank you!

---

> > > ### Comment · Reviewer_cXN6 · 2024-11-25
> > >
> > > Thank you for your responses. I will keep my score as it is.

---

> > > > ### Author Response · Authors · 2024-11-25
> > > >
> > > > Given your stated confidence level of 5 (‘absolutely certain’) and our detailed responses addressing each concern with additional experimental results and clarifications, we would greatly appreciate if you could elaborate on why these responses have not affected your assessment. This would help us better understand the remaining gaps in our work and improve it accordingly.

---

### Official Review · Reviewer_q2eg · 2024-10-29

**Soundness:** 4
**Presentation:** 4
**Contribution:** 3
**Rating:** 6
**Confidence:** 4

**Summary:**

The paper introduces a method for scientific image classification using Visual RAG and external tools with Large Multimodal Models (LLMs)/. For classification, the proposed method tries to mimic an 'expert human' by first retrieving relevant examples from the training to to compare and then using different tools to aid in the classification. They show the effectiveness of their method on 3 real world datasets and also show the reasoning ability through the multi-round dialog component. They also claim that the model is deployed as a web application and is being used by ecologists and scientists.

**Strengths:**

1. A very creative application of a) Visual RAG, b) utilizing external tools and 3) multi-round dialog with LMMs for scientific image classification.

2. The fact that the model is actively deployed and is used by ecologists for aquaculture research is quite impressive.

**Weaknesses:**

1. This kind of framework of using Visual RAG + external tools with Large models is not that novel. However, I do agree that their application in scientific images is quite novel.

2. The method provides reasoning of its predictions through it's transcripts during the multi-round dialog process. Although the transcript makes the model more interpretable and diverse, there is no verification/human evaluation of the correctness of the model's reasoning. A human evaluation study of the correctness of the reasoning would have been great.

**Questions:**

1. The method says that the most similar positive and negative images are retrieved for a test image. However, in the example transcripts, I can't seem to find where this step is happening. Could you explain how this retrieval is happening through an example?

2. The naive baseline of k-nn itself is very strong baseline. Did you find any particular reason for that?

3. It is said that the evaluation are done in two settings: low label (20%) and full labeled (100%). Where are these labels used? to train CLIP+MLP? or to retrieve +ve and -ve examples during Visual RAG?

---

> ### Author Response · Authors · 2024-11-20
> **Review 3**
>
> Thanks for your encouraging feedback and thoughtful questions! We address individual points below:
>
> **Novelty**
>
> Thank you for this observation. We agree, while the individual components of Visual RAG and tool usage with LLMs have been explored separately, our key contribution lies in their integration and adaptation for scientific domains. We demonstrate that this combination is particularly powerful for specialized scientific tasks where interpretability and domain expertise are crucial. The framework's success across multiple scientific domains (aquaculture, eelgrass disease, solar panels) and its real-world deployment with domain experts provides strong validation of this approach. Furthermore, our analysis shows that the synergy between VisRAG and domain-specific tools enables more robust and interpretable scientific image classification than either component alone could achieve.
>
>
> **Human Evaluation**
>
> Thank you for this helpful suggestion - you're right that evaluating the correctness of our model's reasoning would add valuable validation to our work. While our current results demonstrate strong quantitative performance and our domain experts have provided positive qualitative feedback on the reasoning transcripts, a systematic human evaluation would indeed be beneficial. We are currently collecting such evaluations through our deployed application, though this process takes time given the need for expert review. We look forward to sharing these additional insights in future work as they become available.
>
> **Questions:**
>
> 1. *“The method says that the most similar positive and negative images are retrieved for a test image. However, in the example transcripts, I can't seem to find where this step is happening. Could you explain how this retrieval is happening through an example?”*
> In the example transcripts (e.g., C.1), we indicate retrieved images to the model with statements like "This is an example with an aquaculture pond" and "Here is an example without an aquaculture pond." Your point raises an interesting direction for improvement - we could more explicitly explain to the model the purpose of these comparative examples, potentially leading to better performance.
>
>
> 2. *“The naive baseline of k-nn itself is very strong baseline. Did you find any particular reason for that?”*
> The strong kNN performance can be attributed to its use of CLIP embeddings. CLIP is already a powerful model that has learned a rich embedding space, which enables even simple nearest-neighbor approaches to perform well in this space.
>
> 3. *“It is said that the evaluation are done in two settings: low label (20%) and full labeled (100%). Where are these labels used? to train CLIP+MLP? or to retrieve +ve and -ve examples during Visual RAG?”*
> The stated percentages of labeled data (20% and 100%) are used both for training the supervised models and for populating the retrieval database used by the VisRAG component.

---

> > ### Author Response · Authors · 2024-11-25
> >
> > Dear Reviewer,
> >
> > We wanted to thank you for your thoughtful review and let you know we've now added comprehensive results for the reasoning verification study you suggested, along with clearer examples of the retrieval process in action.
> >
> > Your insights have helped strengthen the paper, and we would very much appreciate your feedback on these additions.  If you feel we have adequately addressed your concerns/questions we ask if you would consider raising your score or confidence? Thank you!

---

> > > ### Comment · Reviewer_q2eg · 2024-11-26
> > >
> > > Thank you for clarifying my questions. I would like to keep my positive rating.

---

### Official Review · Reviewer_fK76 · 2024-11-02

**Soundness:** 2
**Presentation:** 3
**Contribution:** 2
**Rating:** 6
**Confidence:** 3

**Summary:**

The paper presents an approach for interpretable image classification tailored to scientific domains. It introduces a framework that enables Large Multimodal Models (LMMs) to perform more accurate and explainable reasoning by integrating Visual Retrieval-Augmented Generation (VisRAG) and domain-specific tool usage. This approach can be effective for niche scientific tasks involving binary image classification, where transparency and understanding of the decision-making process are essential.

**Strengths:**

1.	Ablations to understand importance of VisRAG and Tools
2.	Exact prompts used were provided making reproduceability easier

**Weaknesses:**

1.	Compute heavy multi turn conversation.
2.	Limited to binary classification: The approach in its current form is limited to binary classification. Extending the approach to multiclass classification do not appear to be straightforward. If the authors can possible ways to extend beyond binary classification that would be helpful.
3.	Bias from the prediction of supervised model. As discussed in the supplementary, the framework can be heavily influenced by the supervised model.

**Questions:**

Some suggestions and questions,

1.	It would be interesting to perform an experiment without supervised model in the toolset, considering that the LMM can be heavily influenced by the model.
2.	There is no mention of the train test split for each dataset. It can be relevant information
3.	There is a drop in performance of kNN going from 20% to 100% on Aquaculture dataset. Any explanation for why this is happening?

---

> ### Author Response · Authors · 2024-11-20
> **Review 2**
>
> Thanks for your encouraging review and suggestions for ablation experiments! We address individual points below:
>
> **Computational Cost**
>
> Indeed, we see this as the main trade-off of our method, as inference is significantly more expensive than traditional machine learning methods. However, the benefits of having interpretable inference output justify this cost - a view confirmed by the domain experts we work with.
>
> **Binary Classification**
>
> This is a valuable point - the VisRAG component, as implemented, is built specifically for binary classification. However, as noted by reviewer (BMJ1), an interesting reframing of our method would be to consider VisRAG as just another tool - where the model could request similar examples for any specific class. This approach would enable extension to multi-class settings, which we are exploring in ongoing work.
>
> **Bias from Supervised Model**
>
> This is an important issue, as we pointed out and you note, this was a particular failure case observed for the eelgrass dataset. However, we believe this failure case stems from the fully supervised setting, which underperforms and exhibits more bias than AISciVision. While this is a limitation, it reflects the challenging nature of the dataset for all ML models rather than a specific weakness of our approach.
>
> **Questions:**
>
> 1. *“It would be interesting to perform an experiment without supervised model in the toolset, considering that the LMM can be heavily influenced by the model.”*
> This is a helpful suggestion! - and other reviewers also noted we should study removing the other geo-spatial tools. This is addressed in the general comment above, and we performed this ablation experiment (results can be found in Appendix E of the updated paper PDF).
>
> 2. *“There is no mention of the train test split for each dataset. It can be relevant information”*
> This has been made clearer in the experimental section - thank you for pointing this out.
>
> 3. *“There is a drop in performance of kNN going from 20% to 100% on Aquaculture dataset. Any explanation for why this is happening?”*
> This is an insightful observation that we overlooked - thank you for noticing! Upon further analysis, we believe the kNN performance drop for Aquaculture is due to the choice of CLIP embeddings, which are not optimized for representing satellite images (the CLIP paper authors note this limitation, see Section 3.1.5 of https://arxiv.org/pdf/2103.00020). One possible explanation is that CLIP embeddings don't effectively distinguish between satellite images. In the 100% vs 20% setting, we believe this limitation causes the kNN classifier to retrieve training images that are less similar to the test image. This suggests that effective dataset embeddings are crucial for AISciVision's VisRAG component—an avenue we plan to explore in future work.

---

> > ### Comment · Reviewer_fK76 · 2024-11-23
> >
> > Thanks for additional results and clarifying the queries. I'd like to keep my ratings positive.

---

> > > ### Author Response · Authors · 2024-11-25
> > >
> > > Thank you for your valuable feedback and response here! And if you feel we have adequately addressed your concerns and questions,  we ask if you would consider raising your score or confidence? Thank you!

---

### Official Review · Reviewer_BMJ1 · 2024-11-04

**Soundness:** 2
**Presentation:** 3
**Contribution:** 2
**Rating:** 3
**Confidence:** 4

**Summary:**

The paper introduces AISciVision, a framework designed to enhance the capability of large multimodal models (LMMs) for scientific image classification tasks. The framework comprises two main components: (1) Visual RAG (or VisRAG), which retrieves contextually similar positive and negative labeled images from the training set that serves as an initial prompt to the LMM or VLM model along with the test image that needs to be classified. (2) AI agentic workflow that engages with domain-specific tools in a multi-round conversation to classify input test image. The goal is to accurately classify the image and also generate transparent reasoning transcripts that detail the decision-making process. The paper also introduces a web application for real-time aquaculture pond detection.

**Strengths:**

- The paper is well-written and easy to understand.
- The two-stage process of first using VisRAG followed by an interactive AI agentic workflow with domain-specific tools is innovative and well-integrated.
- The framework demonstrates applicability across different domains and generates reasoning transcripts for enhanced interpretability of image classification.
- The paper introduces a web application for aquaculture pond detection which can serve as a data collection tool to build better algorithms based on user feedback.

**Weaknesses:**

- The paper could benefit from a stronger motivation section that elaborates on the significance of the selected scientific classification tasks and their scientific impact. While generating reasoning transcripts is valuable, the specific implications of correct classification and the benefit of LMMs/VLMs over supervised classification models are still unclear.
- Although the authors claim AISciVision is model-agnostic, experiments are limited to GPT-4o. Including evaluations with other LMMs/VLMs, such as LLaVA [A], BLIP [B], etc. would strengthen this claim and demonstrate broader applicability.
- The paper asserts that it outperforms several fully supervised models; however, Table 1 displays a weak baseline of k-NN as a supervised approach. While CLIP + MLP is also included as a supervised baseline, it performs similarly to, and in some cases better than, AISciVision on two of the three datasets (Eelgrass and Solar). To more effectively justify the advantages of AISciVision, it would be helpful to include stronger and more diverse supervised baselines.
- The Tools for the Aquaculture dataset leverage additional geospatial location metadata and utilize Google Maps APIs to fetch images of areas beyond the input test image. Depending on the selected tool—such as panning left/right/up/down, or zooming out—these tools can access a wider context. This makes it an unfair comparison for baseline models that solely rely on the input images and have not seen the images of the surrounding area.
- The domain-specific tools form a significant part of the AISciVision framework but most of the details are missing in the manuscript and are discussed in the appendix.




**REFERENCES**

[A] Liu, Haotian, et al. "Visual instruction tuning." Advances in neural information processing systems 36 (2024).

[B] Li, Junnan, et al. "Blip: Bootstrapping language-image pre-training for unified vision-language understanding and generation." International conference on machine learning. PMLR, 2022.

**Questions:**

- The inclusion of both positive and negative images as initial prompts is interesting. What specific advantage does this approach offer? How would the model’s performance change if only one positive or negative image is provided?
- Table 2 provides ablations with GPT-4o and different components of AISciVision.In a zero-shot setting, GPT-4o demonstrates strong performance across all datasets. For the Eelgrass and Solar datasets, the majority of performance improvements are attributed to the combination of GPT-4o and VisRAG. What additional benefits do these tools provide in these scenarios?
- In the aquaculture dataset, the most notable improvements come from using GPT-4o along with tools that utilize the Google Maps API and supplemental images. How does the model perform when it is limited to tools similar to those used for the Eelgrass and Solar datasets?
- For the aquaculture dataset, when using images of surrounding areas through the Google Maps API, how do you account for changes in the landscape over time? How does this affect the analysis?
- Why do the initial user prompts to the LMM differ for various datasets when providing positive and negative images?
- Why is the model encouraged to use at least three tools? Does performance vary with fewer or more tools?
- What are the benefits of using VisRAG as a standalone component? Could it be incorporated as one of the tools within the AI agent's workflow?
- How are the domain-specific tools chosen? The same tools are used for two different datasets: the Eelgrass dataset, which contains plant images, and the Solar dataset, made up of satellite images. Why do we refer to them as domain-specific tools?
- The AISciVision framework for Aquaculture pond detection relies on geospatial metadata. What happens if a user does not have the geospatial location metadata for a specific test image while using the web application? Is the framework adaptable for datasets that lack this metadata, or are there alternative features that can be utilized?

**Details Of Ethics Concerns:**

No concerns

---

> ### Author Response · Authors · 2024-11-20
> **Review 1**
>
> Thank you for your thorough review and insightful comments. We address individual points below:
>
> **Motivation/significance of transcript generation for scientific classification**
>
> We are expanding the motivation section in the revised manuscript in conjunction with our ecologist collaborators to better articulate the significance of the selected scientific classification tasks and their broader scientific impact. For example, accurate automated aquaculture pond detection in the Amazon is crucial for monitoring illegal deforestation, water quality impacts from runoff, and land usage for estimating carbon emissions, where misclassification could lead to overlooked environmental violations or wasted conservation resources. Similarly, early detection of eelgrass wasting disease helps prevent devastating coastal ecosystem collapse, as these plants are fundamental to marine habitat health and carbon sequestration.
>
> Further, we will emphasize that generating reasoning transcripts is particularly critical for our use cases. Ecologists working in challenging regions, such as the Amazon, rely on predictions that mimic their own decision-making process, by following a checklist and weighing multiple criteria. These transparent transcripts not only provide a detailed rationale for each prediction but also build trust in the system. Importantly, this framework is actively deployed and in real-world use, where interpretability and trust are essential for adoption.
>
> **Stronger baselines, other LMMs, and fair comparisons**
>
> Thank you for these suggestions! We address these points in detail in the general comment - but we have run ablation experiments to cover all of these settings (2 more LMMs, a Resnet50 baseline, and a same-context comparison), and we believe the results have really helped confirm the necessity of all interoperating components.
>
> **More details on tooling**
>
> We will add a discussion of our domain-specific tools to the main paper, highlighting that they arise from collaboration with domain experts and can be generally applicable to other domains. We are working on fitting that discussion into the main paper. Furthermore, we conduct an ablation experiment with tool choices in Appendix E (see general comment).
>
> **Questions:**
> 1. We found that this approach lets the model compare and contrast both the similar and dissimilar classes, mimicking how a human would approach the task. We now include an ablation experiment testing this (more details in the general comment above).
>
> 2. While GPT-4o with VisRAG alone shows strong performance, the domain-specific tools serve a crucial role in providing interpretability and verification - they allow the model to actively test its hypotheses and generate detailed reasoning transcripts that explain how each tool operation contributed to the final classification decision, making the system more transparent and trustworthy for scientific applications.
>
> 3. We conduct an ablation experiment with tool choices in Appendix E (see general comment) for the aquaculture dataset, comparing geospatial tools with the general image enhancement tools.
>
> 4. This is a great question and, unfortunately, one that is difficult to answer. The timing of the photos can significantly affect the results, as ponds can appear or disappear over time. We are working with domain experts to address this issue by incorporating other non-public datasets, but for now, it remains a source of systemic noise in our dataset.
>
> 5. The prompting was crafted by domain experts to most accurately reflect how they would teach or prompt another human to perform the task - so they are domain-specific.
>
> 6. This is an area we are actively studying. Indeed, selecting three tools was a hyperparameter we set, as we found this provided a good balance between allowing the model to explore more data without overloading it.
>
> 7. This is a really interesting comment. It comes down to system design - we effectively require the model to use the VisRAG tool because we see it as a very powerful component. However, you raise a great point: it would be interesting to study a design where everything is treated as a tool, letting the model freely decide its use.
>
> 8. While both the Eelgrass and Solar datasets leverage similar image enhancement tools (contrast, brightness, edge detection), we classify these as domain-specific because they were selected based on expert consultation and common analysis practices in their respective fields - plant pathology experts commonly use these visual adjustments to identify disease patterns, while remote sensing experts use similar enhancements to detect built structures in satellite imagery.
>
>
> 9. As of now, because we are working specifically with domain experts in the Amazon, they always have latitude/longitude metadata available. For other settings where this metadata is not available, we could craft alternative tooling.

---

> ### Comment · Reviewer_BMJ1 · 2024-11-24
>
> I would like to thank the authors for their detailed responses and for adding ablation studies, which have addressed most of the questions raised in my initial review. However, I have a few additional clarification questions for further discussion:
>
> > C2: Domain-specific tools serve a crucial role in providing interpretability and verification.
>
> Question: Given that GPT-4o + VisRAG contributes most significantly to the performance gain, are the domain-specific tools primarily serving an interpretability role? Additionally, since the LMM outputs already provide interpretability in explaining classification decisions at each turn even after VisRAG, why are the domain-specific tools necessary for interoperability?
>
> > C5: The prompting was crafted by domain experts.
>
> Question: For clarification, the initial prompts differ across datasets. For the Aquaculture and Solar datasets, the initial user prompt is:
> “This is an example of a satellite image with an <dataset_class>. Describe what you see, noting the characteristics that identify it as an <dataset_class>.” Whereas, for the Eelgrass dataset, the initial user prompt is::
> “I will start by first showing you an example image, that is visually similar to the image we want to classify, that does have eelgrass wasting disease ….”
> Could you explain why different prompts were used for these datasets?
>
> > C8: Tools were selected based on expert consultation and common analysis practices.
>
> Question: The domains of eelgrass and solar panels are quite distinct, yet the same set of tools is provided as a choice for both. Could you clarify how these are referred to as “domain expert tools”? Moreover, how do you determine the expected outcomes of these domain-specific tools? Are there studies or evidence showing that domain experts are already employing these kinds of tools in their workflows?
>
> > Additional Questions
>
> 1. AISciVision shows performance gains only on the Aquaculture dataset. For the Solar dataset, CLIP+MLP / ResNet outperforms or matches the AISciVision model across 5 out of 6 metrics. Similarly, for the Eelgrass dataset, the baselines outperform AISciVision on 4 out of 6 metrics, with comparable performance on the other two. If AISciVision does not provide significant classification performance gains, why not position the paper as an interpretable tool, similar to techniques like CAM [C] or Grad-CAM [D]?
> 2. What unique advantage does AISciVision offer over existing interpretability techniques, such as CAM [C] or Grad-CAM[D], which are already well-established for classification tasks?
>
> > References :
>
> [C] Zhou, Bolei, et al. "Learning deep features for discriminative localization." Proceedings of the IEEE conference on computer vision and pattern recognition. 2016.
>
> [D] Selvaraju, Ramprasaath R., et al. "Grad-CAM: visual explanations from deep networks via gradient-based localization." International journal of computer vision 128 (2020): 336-359.

---

> ### Author Response · Authors · 2024-11-25
>
> Thank you for the additional questions! We are glad that our response and new experiments address your initial questions.
> We address individual points below:
>
>
> **Domain-specific tools and interpretability**
>
> We respectfully disagree with the characterization that domain-specific tools primarily serve an interpretability role. Our evidence demonstrates their substantial functional value. The ablation results in Table 2 definitively show that GPT-4o+Tools outperforms GPT4o+VisRAG on the aquaculture dataset in both Accuracy and F1 score, while the full AISciVision framework (GPT4o+VisRAG+Tools) consistently achieves the highest metrics across datasets. This indicates that tools and VisRAG serve distinct, complementary functions rather than tools being merely interpretability aids. The tools enable precise specialized analytical operations that neither the LMM nor VisRAG can perform independently, such as edge detection, and further analysis that are essential for accurate classification. Importantly, these tools don't just explain decisions - they fundamentally shape how decisions are made by enabling the LMM to perform the same analytical steps that domain experts use to reach their conclusions.
>
> The tools serve two equally important functions: enabling accurate classification through domain-specific analysis capabilities while simultaneously providing clear interpretability of the decision-making process.The fusion of VisRAG's broad visual understanding with the tools' specialized analytical functions creates a system that better replicates expert workflows, leading to both better performance and more trustworthy results.
>
>
> **Slight prompt difference in datasets**
>
> We appreciate you bringing this to our attention, and sorry we missed your initial point! The prompt difference between datasets is an unintentional artifact from our prompt refinement process during development, where we failed to standardize the prompts across all datasets in our final implementation. Upon thorough review of our experimental results and the tool-call processes, which use consistent prompting across datasets, we are confident that this prompt variation does not materially impact our reported findings or conclusions. Nevertheless, we acknowledge this inconsistency and have updated our codebase to use standardized prompts across all datasets for better reproducibility and clarity. Thank you for catching this!
>
> **Tools selected by consulting experts**
>
> Thank you for this important question about domain specificity. The tools have varying levels of domain alignment across our datasets. For aquaculture and eelgrass, our tool selection was directly informed by working with our ecologist collaborators who use similar analytical approaches in their workflow. The solar panel dataset was included as an additional test case to evaluate our framework's generalizability to other remote sensing classification tasks, where we just used similar image processing techniques as tools.
>
> While our tool selection shows promise across these different domains, we acknowledge that future work would benefit from deeper domain expert consultation for each specific use case to further optimize tool selection and integration.

---

> > ### Author Response · Authors · 2024-11-25
> >
> > **Similarities and differences to GradCAM-style interpretability methods**
> >
> > We have to respectfully again disagree with positioning AISciVision solely as an interpretability method. The performance patterns across datasets actually reinforce one of our key findings: the effectiveness of AISciVision correlates strongly with the level of domain expert consultation in tool selection and workflow design. In the aquaculture dataset, where we had extensive collaboration with expert ecologists to design domain-specific tools and workflows, AISciVision significantly outperforms baseline methods. This suggests that when properly aligned with domain expertise, our framework can achieve both superior performance and interpretability.
> >
> >
> > Our framework provides several unique advantages beyond traditional interpretability methods like GradCAM. First, AISciVision is a lightweight technique to specialize existing LLMs for niche scientific domains using only prompt design, pretrained image embeddings, and domain-specific tools. This makes it particularly accessible to scientific practitioners. Second, unlike GradCAM-style methods that produce activation maps requiring manual inspection of each example, AISciVision generates programmatically analyzable text justifications. This enables domain experts to perform dataset-wide analyses of prediction rationales, answering high-level questions about classification patterns that wouldn't be possible with traditional visualization methods.
> >
> >
> > Furthermore, AISciVision's LLM-agnostic architecture means it can immediately leverage improvements in foundational models as they emerge. This 'future-proof' design, combined with our finding that performance improves with domain expert involvement, suggests that AISciVision's capabilities will continue to grow as both LLM technology advances and domain-specific implementations are refined. We are also actively exploring how AISciVision can enable scientists to provide verbal feedback on incorrect classifications, creating an interactive learning loop that traditional interpretability methods cannot support.
> >
> >
> > The varying performance across datasets ultimately demonstrates the importance of domain expert consultation in tool selection and workflow design - a finding that helps guide future implementations of our framework.

---

### Author Response · Authors · 2024-11-20
**Rebuttal**

We thank the reviewers for their thoughtful feedback and for engaging deeply with our work. We are particularly pleased that the reviewers found our approach creative (q2eg), well-written (BMJ1), innovative (BMJ1) and an effective application of LMMs to real-world scientific problems (q2eg, fK76, cXN6). The feedback has provided many valuable suggestions for improving both the clarity and comprehensiveness of our work. We have addressed the raised concerns and are excited to share several significant updates:

**Additional Baselines and Models**
* We now include evaluations with additional LMMs/VLMs - Claude-3.5-Sonnet [1] and Llama-3.2-11B [2,3], to demonstrate the model-agnostic nature of AISciVision (Table 4 in Appendix E), as suggested by reviewer BMJ1. Additional experiments followed expected patterns across LMMs: VisRAG and domain-specific tools deliver complementary benefits in performance metrics with any choice of LMM (Table 5 in Appendix E with Llama-3.2-11B). These experiments validate the efficacy of our proposed AISciVision framework.
* We have also added a Resnet50 baseline across all datasets (Table 1), addressing BMJ1's concern about the need for stronger supervised baselines. While Resnet50 performed well, particularly on the eelgrass dataset, AISciVision with GPT-4o outperformed it for aquaculture and solar datasets. Resnet50 particularly struggled because of the high class imbalance and difficulty of these datasets [4] - highlighting the strength of our framework for scientific image classification.
* Reviewer BMJ1 also raised the valid concern that AISciVision might have an unfair advantage on the aquaculture dataset because it accesses additional satellite data through its tools. We view the ability to flexibly access additional data to be a key strength of AISciVision. While we can devise model-specific approaches and training paradigms to select how to query additional data, AISciVision readily handles this in a model-agnostic manner. Nevertheless, it is an interesting question to understand how a standard neural classifier performs when given additional data. To address this, we conducted an experiment where Resnet50 was given the same set of images accessed by our method and made predictions via a majority vote (Table 7 in Appendix E). Even under this fairer comparison, AISciVision outperformed Resnet50, highlighting the robustness of our framework. We acknowledge that accessing additional data could be seen as an advantage, and this is a question we grappled with internally. On one hand, incorporating more context could be viewed as giving our method an edge; on the other hand, the ability to intelligently query and integrate multiple data sources is a fundamental strength of AISciVision, designed to address the complexities of scientific classification. We fully agree with the reviewer’s concern and have now included results under a fairer comparison to address this point.

**Additional Experiments**
We performed several new ablations to further analyze AISciVision's components:
* VisualRAG Ablation: As suggested by BMJ1 and q2eg, we ran experiments returning only a single positive or single negative image in the VisualRAG process (Table 6 in Appendix E). These experiments revealed that using both positive and negative examples, as in our full method, yields the best results, emphasizing the complementary value of both.
* Aquaculture Dataset Without Tools: Addressing fK76's concern about supervised model reliance, we evaluated the aquaculture dataset without the supervised tool and, separately, without satellite tools like Google Maps (Table 8 in Appendix E). Both removing the supervised tool, and removing the satellite tools resulted in a slight performance dip - consistent with our expectation.

**All additional experiments can be found in Appendix E.**

**Anonymized Open Source Code**

We share an anonymized version of our code here: https://anonymous.4open.science/r/AiSciVision-B2B5/

**References**

[1] https://www-cdn.anthropic.com/fed9cc193a14b84131812372d8d5857f8f304c52/Model_Card_Claude_3_Addendum.pdf

[2] Touvron, Hugo, Thibaut Lavril, Gautier Izacard, Xavier Martinet, Marie-Anne Lachaux, Timothée Lacroix, Baptiste Rozière, et al. ‘LLaMA: Open and Efficient Foundation Language Models’. arXiv [Cs.CL], 2023. arXiv. http://arxiv.org/abs/2302.13971.

[3] https://www.llama.com/docs/model-cards-and-prompt-formats/llama3_2/#-llama-3.2-vision-models-(11b/90b)-

[4] Johnson, Justin M., and Taghi M. Khoshgoftaar. "Survey on deep learning with class imbalance." Journal of Big Data (2019)

---

### Meta-Review · Area_Chair_B1v4 · 2024-12-21

**Metareview:**

The AC appreciates the authors' detailed responses and the exploration of applying the AISciVision framework to new domains. However, the paper falls short in several critical areas. First of all, AC questions the usefulness of focusing on classification as the task. AC also recognizes that the domain-specific tools used for the eelgrass and solar panel datasets are identical, undermining the claim of domain specificity. For the aquaculture domain, the reliance on Google Maps APIs to provide additional images does not convincingly demonstrate domain-specificity.  Additional issues include the high risk of bias when comparing new data to similar datasets, the challenges in defining domain-specific tools without sufficient expert input, and the added inference time and cost caused by multiple prediction rounds, etc. etc.  Given these substantial limitations, and AC shares strong negative views with two reviewers, the submission does not meet the standards for acceptance.

**Additional Comments On Reviewer Discussion:**

Reviwers with negative opinions futher firmly stand with their view, and AC agrees.

---

### Decision · Program_Chairs · 2025-01-22

Reject